# Massive remobilization of permafrost carbon during post-glacial warming

T. Tesi[1,2,3], F. Muschitiello[2,4,5,6], R.H. Smittenberg[2,6], M. Jakobsson[2,6,7], J.E. Vonk[8], P. Hill[7], A. Andersson[1,2], N. Kirchner[2,9], R. Noormets[7], O. Dudarev[10,11], I. Semiletov[10,11,12] & Ö Gustafsson[1,2]

Recent hypotheses, based on atmospheric records and models, suggest that permafrost carbon (PF-C) accumulated during the last glaciation may have been an important source for the atmospheric $CO_2$ rise during post-glacial warming. However, direct physical indications for such PF-C release have so far been absent. Here we use the Laptev Sea (Arctic Ocean) as an archive to investigate PF-C destabilization during the last glacial–interglacial period. Our results show evidence for massive supply of PF-C from Siberian soils as a result of severe active layer deepening in response to the warming. Thawing of PF-C must also have brought about an enhanced organic matter respiration and, thus, these findings suggest that PF-C may indeed have been an important source of $CO_2$ across the extensive permafrost domain. The results challenge current paradigms on the post-glacial $CO_2$ rise and, at the same time, serve as a harbinger for possible consequences of the present-day warming of PF-C soils.

[1] Department of Environmental Science and Analytical Chemistry (ACES), Stockholm University, Svante Arrhenius väg 8, SE-11418 Stockholm, Sweden. [2] Bolin Centre for Climate Research, Stockholm University, SE-106 91 Stockholm, Sweden. [3] Institute of Marine Sciences, National Research Council (ISMAR-CNR), Via Piero Gobetti 101, 40129 Bologna, Italy. [4] Lamont-Doehrty Earth Observatory, Columbia University, 61 Route 9W—PO Box 1000, Palisades, New York 10964-8000, USA. [5] Uni Research Climate, Nygårdsgaten 112, 5008 Bergen, Norway. [6] Department of Geological Sciences (IGV), Stockholm University, Svante Arrhenius väg 8, SE-106 91 Stockholm, Sweden. [7] University Centre in Svalbard (UNIS), P O Box 156, N-9171 Longyearbyen, Svalbard. [8] Faculty of Earth and Life Sciences, Department of Earth Sciences, VU University Amsterdam, De Boelelaan 1085, 1081 HV Amsterdam, The Netherlands. [9] Department of Physical Geography (NG), Stockholm University, SE-106 91 Stockholm, Sweden. [10] Pacific Oceanological Institute FEB RAS, Baltic Street, 690041 Vladivostok, Russia. [11] Tomsk Polytechnic University, Lenina Prospect, 634050 Tomsk, Russia. [12] University of Alaska Fairbanks, Koyukuk Drive, Fairbanks, Alaska 99775-7340, USA. Correspondence and requests for materials should be addressed to T.T. (email: tommaso.tesi@ismar.cnr.it).

The Arctic and sub-Arctic soils hold about twice as much carbon as the pre-industrial carbon inventory in the atmosphere[1,2]. Recent studies have suggested that the permafrost carbon (PF-C) stock in permanently frozen soils during the Last Glacial Maximum (LGM) might even have been substantially higher (by 700–1,000 Pg C) than the contemporary stock in high-latitude soils (1,300 Pg C)[2–4]. Most of this megapool of PF-C was formed during the Pleistocene through episodes of sediment deposition, which, combined with the low temperatures, resulted in large burial of terrigenous biomass in soils that were not covered by ice sheets[3,4].

The last glacial–interglacial transition represents a major climatic reorganization during which the Northern Hemisphere became warmer while the atmospheric $CO_2$ rose from ca. 190 parts per million by volume (p.p.m.v.) to ca. 270 p.p.m.v.[5,6] (corresponding to ca. 190 Pg C)[2]. This transition was not gradual, but characterized by abrupt fluctuations in temperature and atmospheric $CO_2$ within the general post-glacial trend. A unique process to explain the $CO_2$ variations has so far not been found. Upwelling of poorly ventilated abyssal water masses is one of the leading hypotheses[7,8], although the search for a $^{14}C$-depleted carbon reservoir in the deep ocean remains a matter of current debate[9,10]. Recently, alternative studies based on atmospheric records[11], terrestrial records[3] and modelling exercises[2,12,13] have inferred that destabilization of PF-C reservoirs might have played a key role in regulating atmospheric $CO_2$ levels on glacial–interglacial timescales. However, despite the large amount of carbon held in the northern soils at the LGM, we still do not have direct physical evidence to understand whether this large carbon reservoir remained dormant throughout the climate transition or whether the thermal reactivation of PF-C resulted in large-scale carbon redistribution between earth system compartments.

In this study, we use Laptev Sea sediments (Arctic Ocean) to investigate the putative instability of PF-C in the Lena River catchment (central Siberia) during the last glacial–interglacial period. Initially, we focus on the Younger Dryas-Preboreal transition (YD-PB, ca. 11,650 yBP) during which the Northern Hemisphere experienced an abrupt temperature increase. The YD cooling event was a sudden return to near-glacial conditions during last deglaciation, likely to be triggered by atmosphere–ocean feedback associated with meltwater fluxes and rapid sea ice expansion combined with a weakened Atlantic Meridional Overturning Circulation[14–16]. Ice cores from Greenland have documented the re-establishment of the post-glacial warming in the $\delta^{18}O$ record[17] as a sudden temperature increase. Given the abrupt nature of the YD termination, the YD-PB climate transition represents an ideal benchmark to assess the stability of PF-C during rapid climate warming. Finally, based on a collection of published studies[18–21], we investigate the PF-C remobilization over a longer deglacial warming period. This study thus presents observation-based evidence of massive PF-C destabilization during past warming events and contributes a new angle to the ongoing debate on mechanisms driving the increase of atmospheric $CO_2$ during the last deglaciation. Our results suggest that thermal reactivation of dormant permafrost might have been an important source of carbon[2,12].

## Results

### Deposition of permafrost in the Laptev Sea during the YD-PB transition.
Because of its extensive catchment ($2.5 \times 10^7$ km², the second largest in the Arctic and sub-Arctic regions), the soil carbon supplied by the Lena River to the Arctic Ocean integrates the signal from a vast permafrost domain, which, during the LGM, was largely dominated by the tundra-steppe biome but

with the presence of woody taxa as well[22–24]. The piston core investigated in detail here (PC23) was collected in the paleo valley of the Lena River (56 m water depth), now in the central-outer Laptev Sea shelf (Fig. 1 and Supplementary Fig. 1). The downcore chronology was established using Bayesian age-depth modelling based on radiocarbon dating of marine molluscs and large vascular plant fragments (Supplementary Fig. 2 and Supplementary Table 1). The resulting age model indicates that the sedimentary record begins shortly before the termination of the YD and encompasses the entire Holocene.

The late Holocene sediment accumulation rate in the inner shelf of the Laptev Sea is on the order of $41 \pm 13$ cm ky$^{-1}$ on the basis of $^{14}C$ dating of fossil calcareous organisms[19]. Similar rates were measured in the Lena prodelta using $^{137}Cs$ and $^{210}Pb$ radioisotopes[25]. Our record shows that during the YD-PB transition the sediment accumulation in the paleo inner shelf was one order of magnitude higher ($337 \pm 27$ cm ky$^{-1}$). Highly laminated sediment strata (Supplementary Fig. 3)—typical of event-driven deposition in river-dominated margins with an absence of bioturbation[26]—corroborate the rapid emplacement of this deposit. Most importantly, our record indicates that the YD-PB transition in the Laptev Sea was characterized by elevated organic carbon (OC) fluxes ($101 \pm 18$ gC m$^{-2}$ y$^{-1}$; Fig. 2b). These figures are particularly high when compared with the late Holocene flux (0–7,000 yBP) in the Lena prodelta (from 0.04 to 25 gC m$^{-2}$ y$^{-1}$)[19].

By using bulk isotopes and molecular proxies as source diagnostic tools, we have found that the high sedimentation rates and carbon fluxes during the YD-PB warming event reflect a massive deposition of land-derived material. Specifically, the OC deposited during this period is characterized by depleted stable carbon isotopic composition ($\delta^{13}C = ca.\ -27‰$) typical of terrestrial vegetation (Fig. 2c)[27]. Analysis of fossil biomarkers offers important additional information, because it allows tracing exclusively land-derived carbon pools, circumventing dilutions by $\delta^{13}C$-depleted river/estuarine phytoplankton, which can exhibit an isotopic signature similar to terrestrial vegetation in the Lena watershed[28]. In this study, we have focused on lignin phenols, a structural biopolymer of higher plants, and cutin-derived products, a waxy biopolymer of plant cuticles[29]. In PC23 fluxes of lignin phenols and cutin-derived products peak at the YD/PB transition consistent with the overall $\delta^{18}O$ trend (Fig. 2f,g). Furthermore, their carbon-normalized concentrations are remarkably consistent with the average values observed in Siberian permafrost soils[30] (Supplementary Fig. 4), further confirming the massive OC deposition of terrestrial origin.

According to the age-depth model, the high land-to-ocean carbon fluxes started before the YD termination (ca. 11,650 yBP). After the cold and dry YD onset, the Northern Hemisphere experienced a second phase of the YD relatively more humid and warmer than the initial conditions. In Europe, this was associated with the resumption of North Atlantic overturning[31]. Ice cores from Greenland have recorded the period before the YD termination as a progressive decrease of the Asian dust concentration (Fig. 2d)[32]. This was likely to be the result of northward migration of the intertropical convergence zone and progressively wetter conditions over the Asian deserts, which would justify relatively high OC fluxes already towards the end of the YD[33]. Climatic reconstruction based on pollen assemblages from the Lena watershed confirms the increase of mean temperature and precipitation towards the end of the YD[24], consistent with the high OC fluxes of terrestrial origin observed in our record.

### Source of the translocated terrigenous carbon.
To quantify the relative input from active layer and deeper/older permafrost sources, we have applied a mixing model based on $\delta^{13}C$, $\Delta^{14}C$

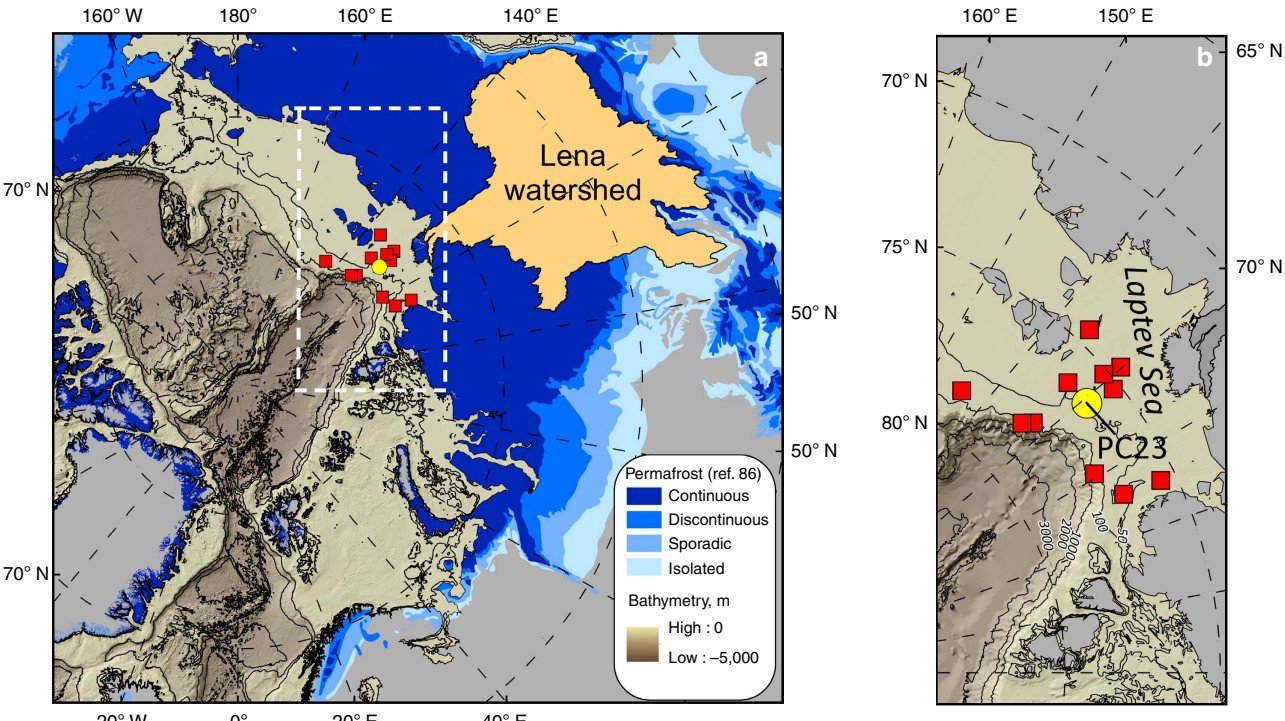

**Figure 1 | Location of post-glacial sediment records in the Laptev Sea. (a)** Blue shades show the extent of permafrost as continuous, discontinuous, sporadic and isolated[86]. The Lena River watershed is shown as an orange area. The location of PC23 collected during the SWERUS-C3 expedition (2014) in the mid-outer shelf of the Laptev Sea is shown as a yellow filled circle. Filled red squares display the location of other transgressive deposits from published studies that were used for carbon deposition and flux calculations[18,20,82]. **(b)** Close-up of the sediment records in the Laptev Sea. The region of **b** corresponds to the open rectangle shown in **a**.

and lignin data using a Monte Carlo simulation to account for the natural variability of the end members[34] (Fig. 3). The statistical mixing model indicates that the material held in the active layer was the major carbon source, accounting for 70–80% of the carbon supplied during the YD-PB transition (Fig. 4). In our model, ice-rich coastal deposits (that is, Pleistocene Ice Complex Deposit (ICD)), which dominated the lowlands of northern Siberia during the LGM (Figs 3 and 4) and which today still stretch along the Laptev Sea and East Siberian Sea coasts, have been considered part of the deep sources[35]. This view is consistent with a significant fraction of this [14]C-depleted Pleistocene deposit today entering the margin via coastal erosion and thermal destabilization processes[36,37]. However, despite the abrupt warming and the post-glacial sea level rise— *ca*. 4 m[38] equivalent to 20–40 km of coastal flooding during the high sediment accumulation period—source apportionment results undoubtedly indicate that PF-C was supplied from the watershed via surface water runoff rather than thermal collapse or erosion of ice-rich coastal deposits.

Vegetation proxies provide additional important evidence regarding the OC source[39] and further confirms the river-derived origin of PF-C. In particular, pollen assemblages from the northern Lena region have shown that trees and shrubs appeared in the northern Siberia (Lena delta region) only after the onset of the Holocene, while during the LGM their abundance progressively increased going southward with some taxa present across their contemporary ranges[22,23]. Lignin fingerprint of ICD samples[30,39] collected in the Northern Siberia confirms their tundra/steppe-like origin (that is, grass vegetation) in the form of a relatively higher abundance of cinnamyl phenols compared with vanillyl phenols (that is, high C/V ratio; Fig. 5). By contrast, the lignin signature of PC23 shows a much greater influence of

angiosperm and gymnosperm woody material (low C/V ratio), which is indicative of shrub and tree sources, further corroborating the watershed origin. As both ICD and PC23 exhibit the same degradation extent based on lignin fingerprint (Supplementary Fig. 5), we conclude that the compositional differences reflect indeed a distinct origin.

In addition, compound-specific hydrogen isotopes ($\delta^2$H) of leaf wax HMW *n*-alkanes (common compounds of plant cuticles) suggest that the watershed-derived material was originally PF-C. Although hydrogen fractionation in plants can occur during evapotranspiration along with plant physiology[40], $\delta^2$H of wax plant lipids largely mirrors the original meteoric signal, which, in first-order approximation, varies as a function of the temperature during moisture formation and condensation[41,42]. Therefore, the depleted $\delta^2$H signature of HMW *n*-alkanes in the terrestrially dominated section of PC23 indicates that the carbon deposited during the YD-PB transition is indeed terrestrial biomass originally photosynthesized during a cold period most likely to be in a permafrost-dominated environment (Fig. 2e, Supplementary Fig. 6 and Supplementary Table 3). The $\delta^2$H values measured in PC23 are in fact consistent with a well-studied permafrost loess-paleosol sequence located *ca*. 1,200 km upstream of the current Lena river mouth, which has documented the isotopic depletion of leaf wax *n*-alkanes during cold periods throughout the last two glacial cycles[3].

**OC translocation to the Laptev Sea.** We have estimated the PF-C buried in the Laptev Sea during the last phase of the deglaciation and early Holocene warm period by compiling all published sediment records available in the region (Fig. 1 and Supplementary Figs 7 and 8), to further explore the deposition of PF-C over a longer timescale characterized by warming

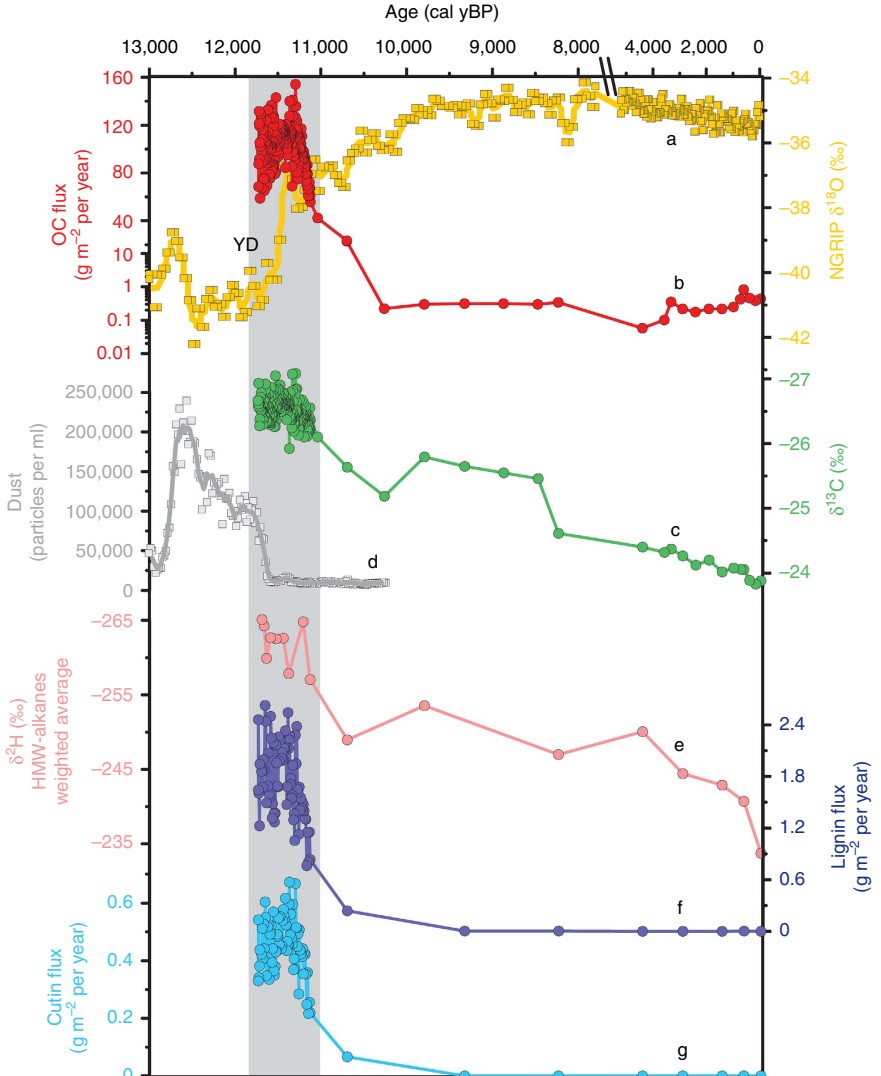

**Figure 2 | Sediment record of massive PF-C remobilization during the YD-PB transition and ice core data from Greenland.** (a) Oxygen isotopes ($\delta^{18}$O) from Greenland ice cores[17] (NGRIP; yellow squares and line, five-point weighted average); (b) OC flux (red circles and line and (c) $\delta^{13}$C (green circles and line) from the sediment record collected in the Laptev Sea (PC23); (d) dust concentration (particles per ml) from Greenland ice cores[32] (NGRIP; grey squares and line, five-point weighted average); (e) weighted average hydrogen isotopic composition ($\delta^2$H) of high-molecular-weight (HMW) saturated odd n-alkanes (C25, C27, C29 and C31; pink circles and line), (f) flux of lignin phenols (dark blue circles and line) and (g) cutin-derived products (light blue circles and line) from PC23 core. The grey vertical region shows the high OC accumulation period. The uncertainty in the Greenland ice core chronology at the YD-PB transition is 99 years (2$\sigma$, not shown)[87].

conditions. The combined data set encompasses 12 sediment cores (Fig. 1), which all demonstrate elevated sediment accumulation rates from the Bølling-Allerød period to the Holocene Climate Optimum[18,20,21]. Accumulation rates are consistent with PC23, which is geographically placed in the middle of the sediment records (Fig. 1). The $\delta^{13}$C data, available for three of the additional records, are depleted as PC23, which confirms the deposition of predominantly terrigenous material, regardless of the core location[43] (Supplementary Fig. 8). The spatial interpolation of these deposits (Supplementary Fig. 9) yields an estimate of 17 ± 6 Pg C (bulk OC) buried in the Laptev Sea during the time period that goes from the mid-Bølling-Allerød to the Holocene Climate Optimum (that is, grey area in Supplementary Fig. 7; from ca. 14 to 7 ky BP). As some of these cores were collected in paleo-channel settings over the mid-shelf, care should be placed on the absolute value of this upscaling estimate, because these records might not be fully representative of the area.

However, as the greatest thicknesses were observed in the outer-shelf—with no evident paleo-channel environments—these latest records have comparatively more importance for the upscaling in comparison with shallower deposits (Supplementary Fig. 10).

By using the source apportionment results of PC23 and accounting for post-depositional degradation (Supplementary Methods, Source apportionment calculations ), we estimate an active-layer carbon cumulative flux to the seabed of 31 ± 9 Pg C from the watershed adjacent to the Laptev Sea, which yields an average annual land-to-ocean export of 4.5 ± 1.4 Tg C y$^{-1}$. Comparison with the modern input from the Lena and other minor rivers (0.66 Tg C y$^{-1}$; Supplementary Methods, Comparison with the modern river input; Supplementary Fig. 11) reveals that PF-C supplied to the Laptev Sea during the late deglaciation/early Holocene was particularly elevated and equivalent to at least seven times the modern river input, which further confirms the massive release of PF-C as a result of the post-glacial warming.

## Discussion

This study provides observation-based evidence of massive PF-C destabilization during the YD-PB transition. Specifically, our data suggest extensive active-layer translocation in the Lena watershed and surrounding catchments adjacent to the Laptev Sea. Collective evidence based on published studies in the region also reveals high land-to-ocean carbon fluxes throughout the last deglaciation[18,20,43,44].

We infer that the enhanced sediment production in response to the warming probably played first-order control on the greater

terrestrial carbon fluxes. Specifically, Siberian lands during cold periods (for example, LGM and YD) are dominated by extensive and continuous permafrost underlying a thin active layer as suggested by paleo-permafrost modelling[45] (Fig. 6a). Under these

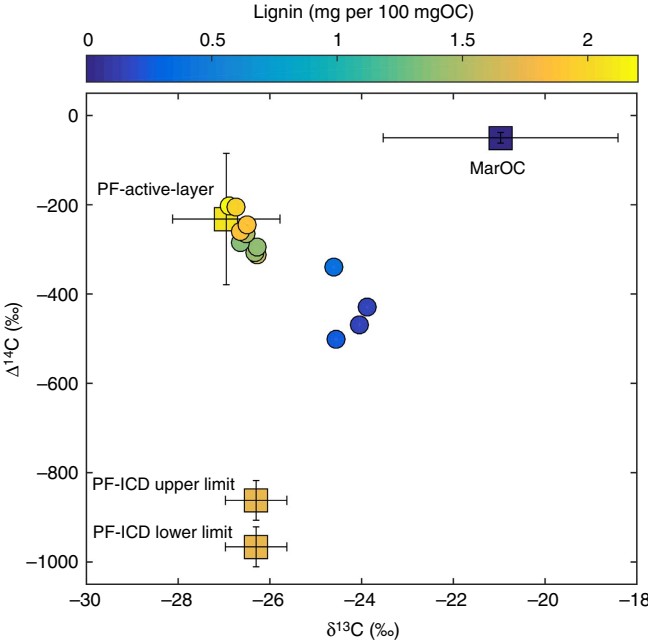

**Figure 3 | OC source apportionment in the Laptev Sea sediment record.** The figure shows the relative contribution of three distinct OC sources shown as squares: marine OC (MarOC), Permafrost ICD (PF-ICD and PF-Active-layer). The source apportionment is based on $\delta^{13}C$, $\Delta^{14}C$ and lignin content (shaded colours).The error bars show the natural uncertainties of each source based on literature data. Further details about the endmember definition and the source of the data are provided in the Supplementary Methods. Circles show the composition of the sedimentary OC in PC23. The radiocarbon signature of the ICD was corrected to account for the $^{14}C$ decay (see Supplementary Methods) based on the age-depth model. The upper and lower limits of PF-ICD show the range of radiocarbon values obtained with the correction.

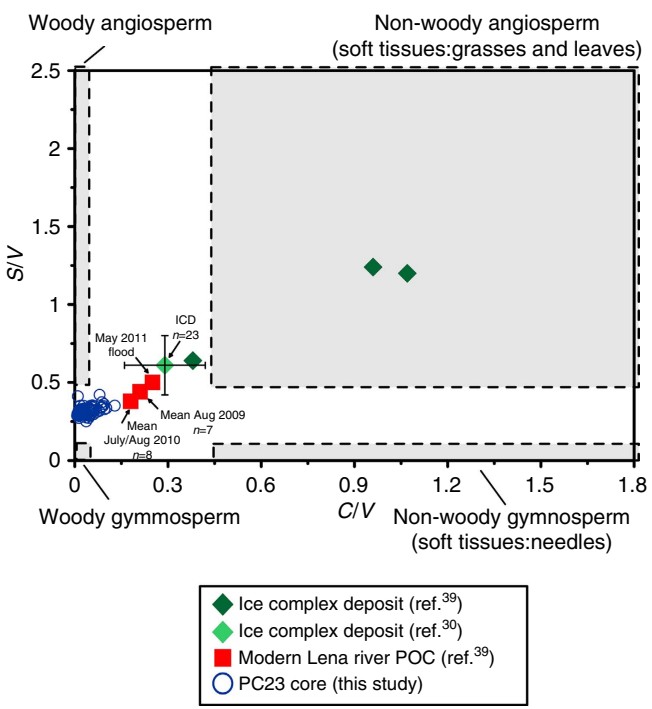

**Figure 5 | Lignin fingerprint of the YD-PB transition in PC23 sediments.** Grey boxes show the plant type based on the typical lignin fingerprint range: woody gymnosperm (bottom left), woody angiosperm (top left), soft-tissue gymnosperm (bottom right) and soft-tissue angiosperm (top right). During the Pleistocene, relatively high syringyl phenols to vanillyl phenols ratios ($C/V$) and high cinnamyl phenols to vanillyl phenols ratios ($S/V$) indicate an important soft-tissue contribution in ICD samples (green diamonds) likely to be in the form of grass-like material typical of tundra/steppe biome[30,39] (error bars display the standard deviation of 23 measurements). Lignin fingerprint of PC23 (blue circles) indicates a much larger contribution of angiosperm and gymnosperm woody plants (shrubs and trees) from the watershed. This is consistent with pollen assemblages, which indicate the occurrence of woody plants in northern Siberia soils (Lena delta) only after the onset of the Holocene[23]. As a reference the figure shows the modern Lena input based (red squares)[39] (error bars are not available for the averaged values).

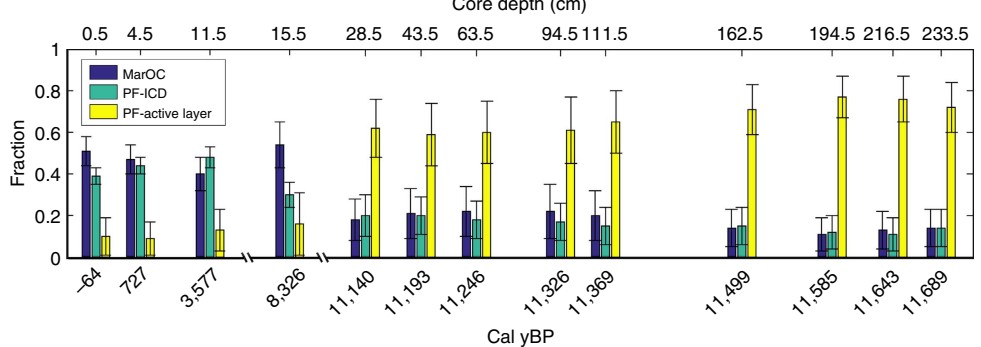

**Figure 4 | Varying contribution of three different carbon source classes through the YD-Holocene.** Results from source apportionment (Markov Chain Monte Carlo simulations) based on $\delta^{13}C$, $\Delta^{14}C$ and lignin concentration. The source classifications were: Marine OC (MarOC; blue), PF-ICD (teal) and PF-Active-layer (yellow). Upper x axis shows the core depth (cm) and lower x axis shows the age (cal yBP). Bars and whiskers display mean and s.d. of the relative contributions from each source based on the Monte Carlo simulations.

 5

conditions, sediment transport models show that the mobility of permanently frozen soils is limited[46]. As the temperature increased during the YD-PB transition[24] (Fig. 6b), the rapid active-layer deepening and the resulting thermokarst development released large quantity of soil carbon[47,48]. In modern Arctic analogues, increased soil erosion as a result of ongoing warming occurs via channelization of previously frozen areas[49,50], active-layer detachment on hillslopes[51], gully developments[52] and retrogressive thaw slump with thawing starting from thermo-karst lakes or stream channels[53] (Fig. 6). It is likely to be that similar processes occurred during past warming events with important implication for the land-to-ocean PF-C fluxes.

On top of this, the termination of the YD corresponds to wetter conditions over the Asian catchments[24], which have been reco-rded in the Greenland ice cores as a drop in dust concentration sourced from the Asian deserts[32,33] (Fig. 2d). This probably enhanced the river runoff and, consequently, the supply of recently destabilized PF-C to the Laptev Sea.

It is also possible that this massive washout of active-layer carbon extended spatially to other nearby Arctic watersheds and shelf seas. For instance, there is strong evidence of high sedimentation during the early Holocene in the Chukchi Sea[54]. The upscaling in the eastern regions is presently challenged by the scarce distribution of data, yet the amount of translocated carbon could readily triple if the East Siberia and Chukchi shelves were included in these estimates. Furthermore, these results only cover part of the deglaciation while similar mechanisms may have also occurred during the earlier stage of the deglaciation with the permafrost thawing front progressively moving northwards in the watershed (Fig. 6b).

The evident implication of this massive land-to-ocean OC flux compared with modern river supply is the elevated $CO_2$ outgassing to the atmosphere, which must have occurred both at site of thawing and during long-range transport through the extensive watersheds and coastal regions. This latter scenario is well illustrated today by the continuous oversaturation of $CO_2$ in the Lena River due to the high reactivity of PF-C[55]. In aquatic environments, the mineralization is particularly efficient when driven by PF-C leaching. Specifically, recent studies have shown that microbial communities in Arctic rivers are capable of rapidly degrading dissolved OC on PF-C thawing regardless of the age of the carbon[56,57]. After deposition on the seabed, early diagenesis of PF-C further continues as illustrated by bottom waters oversaturated in $CO_2$ over terrestrially dominated Laptev sediments[55,58].

However, because of the severe deepening of active-layer permafrost, it is likely to be that most of the OC degradation and $CO_2$ venting occurred at the site of thawing and erosion during the post-glacial warming[11–13,59]. Specifically, the presented shelf record of terrestrial carbon deposition reveals elevated sediment production within the watershed, which implies extensive thawing and, consequently, enhanced microbial respiration of previously frozen PF-C[60,61]. In modern tundra domains, where active layer deepening has been well documented, old carbon loss ranges between 20 and 80 $gC m^{-2} y^{-1}$ (ref. 62). Therefore, given the extent of the Lena watershed and the other rivers adjacent to the Laptev Sea ($3.6 \times 10^7 km^2$ combined), it is likely to be that post-glacial warming resulted in significant $CO_2$ venting from this vast permafrost domain. In fact, collective evidence based on in-situ and laboratory studies indicates that the extent of $CO_2$

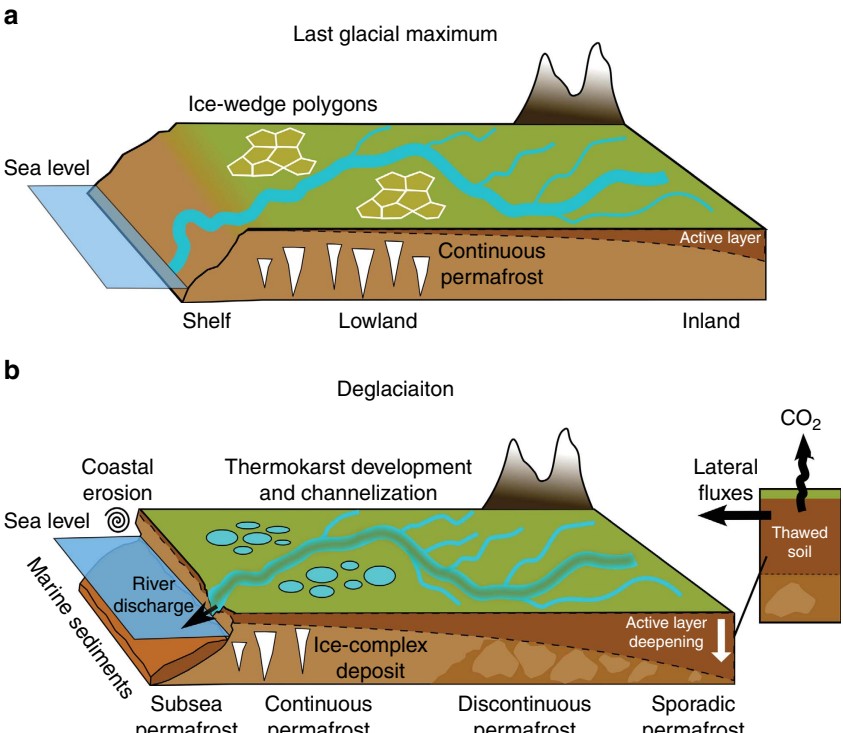

**Figure 6 | Illustration showing the land-to-ocean fluxes during glacial and deglacial periods. (a)** Glacial conditions (for example, LGM) promote rapid accumulation of permafrost carbon[3,4] and result in a thin active layer[45] (dark brown soil colour), while the dominant feature in the watershed is a continuous permafrost (light brown soil colour) with the presence of ice-wedge polygons. Under these conditions, lateral carbon fluxes are small[46]. **(b)** As the temperature increases during the deglaciation, the permafrost conditions change (development of thermokarst lakes and river channelization), while deepening of the active layer (white arrow) releases material previously locked in frozen soils. This ultimately increases the sediment production (horizontal black arrow), which is illustrated by the darker colour of the river network. The destabilization of permafrost soils probably result in large $CO_2$ venting from freshly thawed PF-C (black vertical arrow). Another source of material includes the coastal erosion (black spiral) as the sea level rises.

emissions from PF-C soils appears to be largely related to the amount of the unfrozen OC being released via thawing[63–65].

Regarding the outgassing via anoxic respiration, the $\Delta^{14}C$ signature of the remobilized active-layer permafrost (after correcting for its age[66] and the decay until the end of the YD) is relatively more depleted (ca. 400‰) than the age-corrected $CH_4$ radiocarbon values measured in glacial ice during the YD-PB transition (ca. +150‰)[67]. This further supports the hypothesis that contemporary sources were likely to be responsible for the majority of the YD-PB $CH_4$ rise[67–69]. However, although a contribution from thawing pre-aged PF-C cannot be excluded[70,71], more work is clearly needed to understand the fate of PF-C in response to thermokarst development during Last Glacial Termination.

The increase of atmospheric $CO_2$ by ~80 p.p.m.v. during the deglaciation undoubtedly demonstrates that important reorganization has occurred among Earth system carbon reservoirs[6]. It is still unclear to what extent marine and terrestrial carbon pools have contributed although the emerging picture suggests that a combination of processes must have been operating[2,7,8,10–12]. Our results from Arctic sediments confirm that the last glacial–interglacial transition exerted first-order control on permafrost stability in high latitude watersheds with important consequences for large-scale carbon-climate feedbacks similar to what we might experience as a result of the modern climate change. Given the size of the soil carbon pool held in the northern regions at the LGM[2–4], future paleoclimate modelling of global carbon cycling should include the destabilization of northern soils during the deglaciation and understand the climate-induced redistribution of permafrost carbon among atmospheric, oceanic and terrestrial reservoirs.

## Methods

**Sediment core collection and handling.** The piston core 23 (PC23) was collected in the Laptev Sea (Lat 76° 10.26' N, Long 129° 20.22'E) in July 2014 during the SWERUS-C3 expedition (I/B Oden) at 56 m water depth (Fig. 1 and Supplementary Fig. 1). Split sections were scanned through the ITRAX core scanner (Cox Analytical) to obtain high-resolution X-ray digital images (Supplementary Fig. 2). Step size was set at 340 µm with 20 s exposure time. Post-acquisition image processing was performed with Matlab (Image Toolbox). The downcore sub-millimetre X-ray brightness (greyscale) was first horizontally averaged and subsequently vertically averaged every centimetre. The 1 cm resolved brightness was then fitted with the sediment bulk density to obtain an algorithm to convert X-ray brightness into sediment bulk density (Supplementary Fig. 3).

**Radiocarbon dating and Bayesian age-depth modelling.** For the age-depth model, a selection of 14 samples was sent to the US-NSF National Ocean Sciences Accelerator Mass Spectrometry facility at Woods Hole Oceanographic Institution (Woods Hole, MA, USA) for radiocarbon dating. Before the submission, samples were rinsed with MilliQ water and sonicated for a few seconds. Radiocarbon ages (yBP) are reported following the international convention (Supplementary Table 1)[72,73].

The $^{14}C$ data set of PC23 (Supplementary Table 1) is formed by 11 marine biogenic carbonates covering the uppermost 162 cm and by three large vascular plant debris constraining the lowermost 89 cm (Supplementary Fig. 2a). The radiocarbon data set was modelled using OxCal4.2. The marine $^{14}C$ dates were calibrated using the Marine13 calibration curve[66] whereas the terrestrial $^{14}C$ dates were calibrated using the IntCal13 atmospheric calibration curve[66]. An extra 67 ± 49 (based on the database available on http://calib.qub.ac.uk/marine) and 400 ± 49 $^{14}C$-years regional reservoir effect (ΔR) was added to the top three and the remaining six marine dates, respectively. The larger ΔR (400 ± 49 $^{14}C$-years) in the lower region of the core was imposed because of the depleted $\delta^{13}C$ signature of the biogenic carbonates (− 3.7 ± 1.5‰; Supplementary Fig. 2b) in comparison with the typical signature (~ +1‰) in modern high-salinity waters[18,44]. The lower $\delta^{13}C$ signature is likely to be the result of the freshening during the deglaciation, which implies the supply of a relatively $^{14}C$-depleted carbon source via rivers[44].

Bayesian age-depth modelling with OxCal relies on the correct choice of k, a parameter that determines the degree of flexibility of the model on the $^{14}C$-dating sequence and relates to the nature of the depositional environment[74]. We performed several runs using different k parameters until satisfactory agreement indices[75] were obtained. The sharp lithological boundary at 23 cm, which marks the interruption in deposition of laminated sediments (based on X-ray images; Supplementary Fig. 3 and core description), was also prescribed in the sequence,

to define the step change in depositional rates. The Outlier_Model analysis was performed with the General setting and the prior probability fixed to 0.05, which weighs down the radiocarbon measurement that have statistical probability of > 5% of being outliers[75].

The final model was run using a k-value of 3. This value is suitable to constrain $^{14}C$ data rigidly where little change in sedimentation rate occurred such as for the laminated bottom unit[76]. The output resulted in a robust and coherent age model with an overall solid structure of the dated sequence (Supplementary Fig. 2a) as defined by an excellent agreement index of 94.6% (ref. 75).

**Bulk OC analyses.** OC and stable carbon isotope ($\delta^{13}C$) analyses were carried out on acidified samples (Ag capsules, HCl, 1.5 M) to remove the carbonate fraction[77]. Analyses were performed using a Thermo Electron mass spectrometer directly coupled to a Carlo Erba NC2500 Elemental Analyzer via a Conflo III (Stable Isotope Laboratory (SIL), Department of Geological Sciences, Stockholm University). OC and TN values are reported as weight per cent (% d.w.; Supplemetary Fig. 4a), whereas stable isotope data are reported in the conventional delta notation (‰; Supplementary Fig. 4a). The analytical error for $\delta^{13}C$ was lower than ± 0.1‰ based on triplicate analyses of the same sample.

A subset of 12 bulk sediment samples were acidified (HCl, 1.5 M) and sent to National Ocean Sciences Accelerator Mass Spectrometry for radiocarbon analyses. Radiocarbon data of bulk OC are reported in the standard $\Delta^{14}C$ notation (‰)[73] and as calibrated yBP (Supplementary Table 2).

**Biomarkers.** The alkaline CuO oxidation was carried out using an UltraWAVE Milestone microwave[78]. Briefly, ~2 mg of OC was oxidized using CuO under alkaline (2 N NaOH) and oxygen-free conditions at 150 °C for 90 min in teflon tubes. After the oxidation, known amounts of trans-cinnamic acid and ethylvanillin were added to the solution as recovery standards. The aqueous solutions were then acidified to pH 1 with concentrated HCl and extracted with ethyl acetate. Extracts were dried and redissolved in pyridine. CuO oxidation products were quantified by gas chromatography–mass spectrometry (GC–MS) in full scan mode (50–650 m/z). Before GC analyses, the CuO oxidation products were derivatized with bis(trimethylsilyl) trifluoroacetamide + 1% trimethylchlorosilane at 60 °C for 30 min. The compounds were separated chromatographically in a 30 m × 250 µm DB5 ms (0.25 µm-thick film) capillary GC column, using an initial temperature of 100 °C, a temperature ramp of 4 °C min$^{-1}$ and a final temperature of 300 °C. Lignin phenols and 3,5-dihydroxybenzoic acid were quantified using the response factors of commercially available standards (Sigma-Aldrich), whereas cutin-derived products were quantified by comparing the response factor of trans-cinnamic acid. Lignin-derived reaction products include vanillyl phenols (V = vanillin, acetovanillone, vanillic acid), syringyl phenols (S = syringealdehyde, acetosyringone, syringic acid) and cinnamyl phenols (C = p-coumaric acid, ferulic acid). The quantification of cutin acids focused on the most abundant C16 to C18 hydroxy fatty acids: 16-hydroxyhexadecanoic acid, hexadecan-1,16-dioic acid, 18-hydroxyoctadec-9-enoic acid, 7 or 8-dihydroxy C16 α,ω-dioic acid and 8 or 9, or 10,16-dihydroxy C16 acids[79,80]. Quantification of cutin acids was performed using the response factors of trans-cinnamic acid[17]. Results were presented as fluxes (Fig. 2) and OC-normalized data (Supplementary Fig. 4b).

High-molecular weight wax lipids were extracted using an ASE 200 accelerated solvent extractor (Dionex Corporation, USA) using DCM/MeOH (9:1 v/v) at 80 °C (5 × 10^6 Pa)[81]. Samples were placed in solvent-rinsed stainless steel vessels. The empty space left in the vessels was filled with pre-combusted and solvent-rinsed glass bits. After the extraction, solvent-rinsed activated copper was added to the extracts to remove sulfur. After 24 h, extracts were filtered on pre-combusted glass wool and concentrated with the rotary evaporator. Extracts were then transferred into test tubes, evaporated to complete dryness and re-dissolved in 500 µl of DCM. Extracts were then separated into acid, hydrocarbon and polar fractions using amino-propyl Bond Elut (500 mg per 3 ml) and column chromatography over $Al_2O_3$. Saturated n-alkanes were further isolated using 10% AgNO_3-coated silica gel and quantified by GC–MS in full scan mode (50–650 m/z). The GC was equipped with a 30 m × 250 µm DB5 ms (0.25 µm thick film) capillary GC column. Quantification was performed using the response factors of commercially available standards (Sigma-Aldrich). Initial GC oven temperature was set at 60 °C followed by a 10 °C min$^{-1}$ ramp until a final temperature of 310 °C (hold time 10 min).

**Compound-specific isotope analyses.** The hydrogen-isotopic abundances of saturated n-alkanes were determined in continuous-flow MS. Purified extracts were concentrated and injected (1–2 µl) into a Thermo Trace Ultra GC equipped with a 30 m × 250 µm HP5 (0.25 µm-thick film) capillary GC column. Oven conditions were similar to the setting used for the quantification. The conversion of organic biomarkers to elemental hydrogen was accomplished by pyrolysis at 1,420 °C (Thermo GC Isolink). After the pyrolysis, $H_2$ was introduced into an isotope ratio mass spectrometer (Thermo Scientific Delta VIRMS) for compound-specific determination of $\delta^2H$ values via a Thermo Conflo IV. Only peaks with amplitude (mass 2) between 1,500 and 8,000 mV were used for the evaluation. The $\delta^2H$ values of the n-alkanes were calibrated against the reference substance mix A4 (Biogeochemical Laboratories, Indiana University). The H3 + factor was determined once a day and stayed constant throughout the evaluation. Each sample

was injected three times and $\delta^2H$ values are reported as mean, s.d. and weighted average (Supplementary Table 3 and Fig. 6). Only HMW saturated odd *n*-alkanes (C25, C27, C29 and C31) were quantified. Average s.d. for all evaluated peaks was 2.9‰ (min 0.3‰ and max 6.0‰; Supplementary Table 3).

**OC burial and Lena river input.** The cumulative OC buried in the transgressive deposit in the Laptev Sea was assessed based on published sediment records available on www.pangea.de[18,20,82]. Location, thickness, average OC content and bulk density along are provided in the Supplementary Methods and Supplementary Table 4 along with interpolation methods used for the estimate (OC burial and flux during last phase of the deglaciation/early Holocene).

The modern flux of OC supplied to the Laptev Sea from the Lena River is based on a power law rating curve[83,84] that was built using data collected by the Arctic Great Rivers Observatory (http://www.arcticgreatrivers.org/data.html). See the Supplementary Methods for further details (comparison with the modern river input).

**OC source apportionment.** Dual-carbon isotope- and biomarker-based source apportionment calculations were performed using the average composition of representative end-members (that is, ICD, active layer permafrost and marine phytoplankton). The end-member composition relies on data existing in literature (Supplementary Methods, Source apportionment calculations)

To account for the end-member natural variability, we ran a Markov Chain Monte Carlo simulation to estimate the resulting uncertainties in the calculated fractions[85]. Further details about end-member definition and Markov Chain Monte Carlo are provided in the Supplementary Methods (Source apportionment calculations; Supplementary Table 5).

**Data availability.** All data presented in this study will be publicly available in Stockholm University's Bolin Centre Database (http://bolin.su.se/data/).

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

## Acknowledgements

We thank the Oden Swedish icebreaker crew and the Swedish Polar Research Secretariat staff. We also acknowledge Dr Henry Holmstrand (Delta Facility, Stockholm University) who provided support for compound specific analyses and Kirsi Keskitalo who helped in the lab. This study was supported by the Knut and Alice Wallenberg Foundation (KAW contract 2011.0027), the Swedish Research Council (VR contract 621-2004-4039 and 621-2007-4631) and the Nordic Council of Ministers Cryosphere-Climate-Carbon Initiative (project Defrost, contract 23001). T.T. additionally acknowledges EU financial support as Marie Curie fellow (contract number PIEF-GA-2011-300259). I.S. acknowledges financial support from the Russian Government (grant number 14, Z50.31.0012/03.19.2014) and the Russian Foundation for Basic Research (numbers 13-05-12028 and 13-05-12041), and O.D. from the Russian Scientific Foundation (grant number 15-17-20032). This is ISMAR-BO contribution number 1902. We thank three anonymous reviewers for their constructive comments. Ö.G. acknowledges the European Research Council (ERC-AdG project CC-TOP #695331).

## Author contributions

T.T. and Ö.G. planned the sampling activity. J.E.V. and P.H. provided support in the field together with the Swedish Polar Research Secretariat and the Oden I/B crew. M.J., N.K. and R.N. took care of seismic data acquisition and post-processing. F.M. developed the age-depth model of PC23. T.T. carried out all analytical analyses on PC23. A.A. ran the MCMC simulation for the OC source apportionment. T.T. wrote the paper with input from all co-authors and produced the figures.

## Additional information

**Competing financial interests:** The authors declare no competing financial interests.

