## [Peer Review File · Nature Communications]

Reviewers' comments:

Reviewer #1 (Remarks to the Author):

Review of

Authors: Tesi et al:

Title: Massive remobilization of permafrost carbon during post-glacial warming

submitted to Nature Communications

Date: 25 Feb 2016

This paper compiles sediment core data from the Laptev Sea over the last 12000 years in order to determine if and how much carbon from thawing permafrost during the rapid warming at the end of the Younger Dryas / beginning of the Preboreal (YD/PB) might have been released. The data are focused on one sediment core, but a compilation of other already published data is presented and used for wider interpretation. Measured variables contain organic carbon, ^{13}C , ^{14}C , lignin, cutin, $\text{d}2\text{H}$ on alkanes. The case is made that during the YD/PB warming massive OC released from permafrost warming enters the ocean, and might subsequently also as CO_2 be released to the atmosphere.

These data-based evidences on permafrost changes during a rapid warming event are to my knowledge new and exciting and worth publication in this journal. So far, most evidences on such changes have been indirect (e.g. via atmospheric records or modelling), while here direct data are presented.

While I can not comment on the data methods etc I like to focus on the interpretation and I have a few main comments, which needs clarification:

1. It is argued in line 105 that the study finds high OC fluxes during YD-PB transition, but in Fig 2 organic carbon (OC) fluxes are already high in the cold YD. Thus, the grey box in Fig 2 contains about 800 years of the YD/PB transition, but it is not clear to me, why OC fluxes are already high in the cold YD. The outline of the paper is along the line, that warming induces permafrost thawing and therefore leads to higher OC fluxes to the Laptev Sea. Unfortunately the records end in the YD, so nothing can be said about the changes during the rapid cooling into the YD around 12.6 ka. The authors need to present some ideas on the high OC flux in the YD and maybe also revise their storyline accordingly.

2. Modelling studies on potential future permafrost thawing (e.g. Koven et al. 2011. PNAS,108, 14769-14774. Schuur et al. 2013. Clim. Change 119, 359-374) show that both CO_2 and CH_4 might be released from such an event, with only a few percentages being released as CH_4 . Saying this permafrost thawing might have contributed to the rapid rise in CH_4 that also occurred at the YD/PB transition (e.g. see Monnin et al 2001, your ref 6). Measurements of ^{14}C on CH_4 (Petrenko et al., 2009 Science 324, 506-508), however, have not indicated that the CH_4 rise at the YD/PB transition has been largely influenced by ^{14}C -dead CH_4 , thus a significant contribution from ^{14}C -poor permafrost thawing to the CH_4 rise seems unlikely here. Some discussion in that direction is necessary. Are the fluxes from the Laptev Sea too small to be relevant for such effects on global CH_4 ? For answering / discussing this CH_4 -related issue the authors might also use ice core studies on CH_4 source attributions, e.g. stable isotopes (Möller et al., 2013 NG doi: 10.1038/ngeo1922) or others, (e.g. Baumgartner et al 2012, BG, doi: 10.5194/bg-9-3961-2012).

3. I found the source attribution by using ^{13}C and ^{14}C data most important. Thus, Fig S6 and methods describing this in detail should be contained in the main text.

4. Along these lines: Please make use of the format available in this journal, you have space for 70

refs, 10 figs, 5000 word main text + 3000 word methods (that was the case I last checked some times ago, maybe the editor can specify these numbers). Therefore a substantial part of the methods might be shifted to the main text. For figures, I leave it to the authors, but as said already Fig S6 is certainly worth being shown upfront.

Minors:

- line 172: „17 Pg bulk OC": Is this 17 Pg of C in OC?
- line 175: „cumulative flux to the seabed of 31 PgC". Over which time period?
- line 176, same: the land oce export of 4.5 TgC/yr: in which time period?
- line 204: CO₂ rise during deglaciation is 80 ppmv (not 90 ppmv as stated) for Termination 1 (this is different/higher for other terminations, but since this paper is on the end of Termination 1 you should adjust this number.

Reviewer #2 (Remarks to the Author):

This manuscript addresses a really interesting topic and puts forward a thought-provoking argument for permafrost carbon destabilization as an important source of atmospheric carbon dioxide during post-glacial warming. Overall, I find the data put forward to be of a high quality and the manuscript is certainly going to stimulate a lot of discussion and I therefore believe will be of high enough impact for Nature Communications. The manuscript is well-written, clearly organized, in the appropriate format and all the data put forward is well supported. That said I have some major concerns that need addressing prior to potential publication.

Firstly, everything is really based off of one piston core, I am well aware of the amount of work that goes into this kind of study but I have serious concerns about how representative data from one core is for scaling up from - can the authors provide any further data to back up the representative nature of this core or other similar data from other cores? Secondly, in lines 190-196 the authors describe how a lot of the CO₂ flux from the thawed permafrost would have happened from soils and throughout the fluvial network (the latter has been the focus of recent papers, e.g. Mann et al., 2015: doi:10.1038/ncomms8856; Drake et al., 2015: doi: 10.1073/pnas.1511705112) and this has been shown to be driven through leaching to DOC and then microbial utilization with likely significant outgassing in the aquatic pathway. This should be included in here to make it clear how much C was on the move at this time as what was thawed and buried in the core represents the fraction preserved (i.e. a small proportion of what was likely thawed). Finally, in lines 199-203 the authors provide literature data for how much permafrost carbon can be mobilized and the size of the Lena watershed but do not provide an estimate for how much C could have been mobilized - this is a critical proof of concept that should be included. Basically, how much C could have been mobilized and then firstly for the Lena and then other Siberian watersheds how significant is the total C mobilized in comparison to the increase of atmospheric CO₂ (90ppmv) - basically can you show that enough C was mobilized to drive what is hypothesized?

Reviewer #3 (Remarks to the Author):

Tesi and co-authors present data from a new sediment core retrieved from the Laptev Shelf, more specifically from a paleochannel of river Lena at around 50 m water depth (unfortunately I could not find information in the manuscript on the exact water depth from where the core was raised). The sediment record covers approximately the past 12 kyr and is characterized by a drastic change in sedimentation rate from very high rates before approximately 11.5 kyr to moderately high rates afterwards. Next to the radiocarbon data used for the age model, the data obtained for this core comprise organic carbon contents, $\delta^{13}\text{C}$ of organic carbon, δD of long-chain n-alkanes, lignin and cutin concentrations, as well as $\Delta^{14}\text{C}$ of total organic carbon. The authors interpret their

data, and a compilation of published data from the region, as records of organic matter remobilization from thawing permafrost deposits and suggest that the process strongly impacted atmospheric CO₂ levels.

This is a timely study investigating possible sources of CO₂ released during the deglaciation. Thawing terrestrial permafrost deposits have recently gained increasing attention due to their potential to explain part of the deglacial rise in atmospheric greenhouse gases, as well as due to the potential effect resulting from anthropogenic climate change causing Arctic warming. The records presented here are of good quality and clearly indicate deglacial accumulation of terrigenous organic matter. However, the finding of strongly increased accumulation of terrigenous organic matter in sediments of the Laptev Shelf and slope, namely in paleochannels of the rivers, is not new. There are several publications discussing these deposits in context with deglacial sea-level rise (e.g., Bauch et al., 2001, cited in this study). In contrast, Tesi et al. argue, based on a mixing model using $\delta^{13}\text{C}$, $\Delta^{14}\text{C}$ and lignin concentrations, that "undoubtedly... (permafrost-carbon) was supplied from the watershed via surface water runoff rather than thermal collapse or erosion of ice-rich deposits."

The contrasting view of Tesi et al. relative to that of the authors of previous studies, namely that the authors of this study consider coastal erosion of ice-rich yedoma type deposits during flooding of previously exposed shelves a less important process, is backed up with the results of a mixing model relying on end-members defined from modern observations. While care is taken to correct $\Delta^{14}\text{C}$ of organic matter samples from the sediment core for radioactive decay that occurred since deposition, the end-member $\Delta^{14}\text{C}$ values apparently have not been corrected for decay. This would, however, be necessary for the ice-complex end-member, which today represents a glacial relict deposit and during the last glacial and deglaciation likely has formed the majority of deposits in the study area. The end-member defined for top-soil relies on (arguably scarce) data from modern active layer soils in the Lena catchment and delta, which today consists of deposits fundamentally different from those probably prevalent in the region during the glacial, i.e., the ice-complex deposits. Moreover, I feel that the authors do not fully exploit the potential of their data. They have obtained lignin phenol concentrations for the core. Concentration ratios of individual phenols can be used to apportion vegetation and plant tissue types, which are expected to differ between the southern reaches of the Lena catchment and the northern tundra. Yet, only lignin concentrations are presented (in mg/g OC, which is an uncommon unit in the literature dealing with lignin; in my view the more common unit should be used to assure easy comparison with other published data, i.e., $\lambda\text{-}8$ expressed in mg lignin phenols/100 mg OC). It would be very interesting to see whether the lignin phenol signature supports the assertion of large amounts of organic matter being sourced "from the watershed", which I would assume to encompass also the southern parts of the catchment, where the deglacial warming and permafrost thawing must have been strongest.

Tesi and co-authors furthermore argue that the stable hydrogen isotopic composition of n-alkanes in their core supports an origin from the catchment of the river, as the δD values in the rapidly accumulating late deglacial sediments are similar to values found in a loess deposit 1200 km upstream from the current Lena Delta. They state that the hydrogen isotopes indicate a source from plants growing under cold climate. However, no data exist on the δD signature of alkanes from ice-complex deposits, which were likely prevalent on the flooded shelf. It is conceivable that those alkanes would also carry a δD signature indicative of cold conditions. So the δD data alone do not support an exclusion of coastal erosion of deposits on the flooded shelf as source for the material accumulated in the channel.

In the second part of the manuscript, an attempt is made to estimate the amount of carbon mobilization during deglacial permafrost thaw and sea-level rise. This is done by extrapolating the data from the core studied here and from other published records to the entire Laptev Shelf area. However, considering that the cores were predominantly and deliberately taken in paleochannels of the rivers, as these locations offer high-resolution sediment sequences resulting from

predominant deposition in these paleo-depressions, upscaling accumulation rates found in these settings must result in vast overestimations.

Lastly, it would be desirable to provide the exact results of the end-member modelling, perhaps in a table in the supplement.

**Author responses to review comments on Nature Comm. ms NCOMMS-16-03685
"Massive remobilization of permafrost carbon during post-glacial warming"**

Reviewer#1

This paper compiles sediment core data from the Laptev Sea over the last 12000 years in order to determine if and how much carbon from thawing permafrost during the rapid warming at the end of the Younger Dryas / beginning of the Preboreal (YD/PB) might have been released. The data are focused on one sediment core, but a compilation of other already published data is presented and used for wider interpretation. Measured variables contain organic carbon, ^{13}C , ^{14}C , lignin, cutin, $d2H$ on alkanes. The case is made that during the YD/PB warming massive OC released from permafrost warming enters the ocean, and might subsequently also as CO_2 be released to the atmosphere.

These data-based evidences on permafrost changes during a rapid warming event are to my knowledge new and exciting and worth publication in this journal. So far, most evidences on such changes have been indirect (e.g. via atmospheric records or modelling), while here direct data are presented.

While I can not comment on the data methods etc I like to focus on the interpretation and I have a few main comments, which needs clarification:

1. It is argued in line 105 that the study finds high OC fluxes during YD-PB transition, but in Fig 2 organic carbon (OC) fluxes are already high in the cold YD. Thus, the grey box in Fig 2 contains about 800 years of the YD/PB transition, but it is not clear to me, why OC fluxes are already high in the cold YD. The outline of the paper is along the line, that warming induces permafrost thawing and therefore leads to higher OC fluxes to the Laptev Sea. Unfortunately the records end in the YD, so nothing can be said about the changes during the rapid cooling into the YD around 12.6 ka. The authors need to present some ideas on the high OC flux in the YD and maybe also revise their storyline accordingly.

To address this point it is useful to consider the extent of the Lena river catchment which encompasses 20 latitudinal degrees. This implies large climate gradients within such an extensive watershed. Specifically, after the dry and cold onset of the YD, the moisture over central Asia progressively increased with time. Ice cores from Greenland have recorded this period as a progressive decrease of the Asian dust (Ruth et al., 2003) (see our Fig. 2e). The drop in dust deposition likely involves the migration of the intertropical convergence zone and the progressively wetter conditions over the Asian deserts where the climate was

relatively warmer and wetter compared to the initial phase of the YD (Steffensen et al., 2008). Similarly, other authors have reported that the second phase of the YD as cold in Greenland but humid in Europe which was likely related to the resumption of North Atlantic overturning (Bartolomé et al., 2015). Under this two-phase scenario of the YD, we can expect a progressive increase of the land-to-ocean C flux from the Lena watershed since the abrupt cold YD onset. This would explain why the OC fluxes are relatively high just before the YD/Holocene transition.

However, it is important to highlight that in Fig. 2 the flux of lignin (and cutin) follows the general $\delta^{18}O$ pattern better than bulk OC. The peak of lignin is indeed consistent with the onset of the Holocene and the end of the YD. This is likely because terrestrial biomarkers can trace better the land-derived input than simply bulk OC. We have further elaborated the text to include the aforementioned comments and interpretations (line 114-120 & 126-128).

2. Modelling studies on potential future permafrost thawing (e.g. Koven et al. 2011. PNAS, 108, 14769-14774. Schuur et al. 2013. Clim. Change 119, 359-374) show that both CO₂ and CH₄ might be released from such an event, with only a few percentages being released as CH₄. Saying this permafrost thawing might have contributed to the rapid rise in CH₄ that also occurred at the YD/PB transition (e.g. see Monnin et al 2001, your ref 6). Measurements of $\delta^{13}C$ on CH₄ (Petrenko et al., 2009 Science 324, 506-508), however, have not indicated that the CH₄ rise at the YD/PB transition has been largely influenced by $\delta^{13}C$ -dead CH₄, thus a significant contribution from $\delta^{13}C$ -poor permafrost thawing to the CH₄ rise seems unlikely here. Some discussion in that direction is necessary. Are the fluxes from the Laptev Sea too small to be relevant for such effects on global CH₄? For answering / discussing this CH₄-related issue the authors might also use ice core studies on CH₄ source attributions, e.g. stable isotopes (Möller et al., 2013 NG doi: 10.1038/ngeo1922) or others, (e.g. Baumgartner et al 2012, BG, doi: 10.5194/bg-9-3961-2012).

The CH₄ rise since the LGM is an intriguing topic which is a matter of much current debate, especially the abrupt increase observed at the end of the YD. Suggestions in the literature range from the rapid expansion of wetlands in the tropics to similar mechanisms in northern latitudes (Brook et al., 2000; Smith et al., 2004).

A wet environment (i.e. wetland) is the key prerequisite to develop anaerobic degradation and, consequently, methane production. Indeed, recent studies focused on ongoing climate change indicate that the increase of high-latitude CH₄ emissions will be more related to changes in moisture rather than to increased availability of unfrozen organic matter following permafrost carbon thaw (Olefeldt et al., 2013). In addition to modelling studies mentioned by the reviewer, recent incubation studies have looked at aerobic and anaerobic permafrost mineralization and found that, although CH₄ has higher global warming potential than CO₂, the total C release under anaerobic conditions is substantially reduced (Knoblauch et al., 2013; Lee et al., 2012) especially when compared with CH₄ fluxes from wetlands. By contrast, it is well known that PF-C experiences rapid CO₂ production upon thaw in oxic environments and drainage basins (Drake et al., 2015; Lee et al., 2012; Mann et al., 2015; Schuur et al., 2009; Xue et al., 2016) and in general the amount of C released is strongly correlated to C concentration of PF-C being thawed (Dutta et al., 2006).

Adding on the reviewer's comment, the signature of the PF-C remobilized at the end of the YD is not far off from the radiocarbon value suggested by Petrenko et al 2009 (between ca. -100 and -200‰). Therefore we cannot completely exclude the potential contribution by deepening active-layer PF-C. However, as explained above, the extent of thawing itself does not necessarily translate into high methane fluxes while the CO₂ production does. Given our limited knowledge of soil hydrology in the study area during the YD-PB transition, we included the CH₄ discussion in the main text suggesting the potential contribution from thawing PF-C but, at the same time, we stressed the fact that further work is vital to test whether the right environmental conditions were attained to justify significant CH₄ emissions (line 226-236). For these reasons, we here take a conservative approach and stay away in this ms from a speculative rough estimation of the atmospheric evasion of GHG that to some extent was likely co-occurring with the massive horizontal remobilization of carbon from the permafrost.

3. I found the source attribution by using 13C and 14C data most important. Thus, Fig S6 and methods describing this in detail should be contained in the main text.

Figure S6 was moved to the main text as suggested. This is now new Fig. 3

4. Along these lines: Please make use of the format available in this journal, you have space for 70 refs, 10 figs, 5000 word main text + 3000 word methods (that was the case I last checked some times ago, maybe the editor can specify these numbers). Therefore a substantial part of the methods might be shifted to the main text. For figures, I leave it to the authors, but as said already Fig S6 is certainly worth being shown upfront.

Overall, the text was further expanded, new references added, and the Fig S6 is now part of the main paper (new Fig. 3)

Minors:

- line 172: „17 Pg bulk OC”: Is this 17 Pg of C in OC?

Yes it is. This part was replaced with “17±6 Pg C (bulk OC)” (line 184)

- line 175: „cumulative flux to the seabed of 31 PgC”. Over which time period?

Calculations regarding flux/burial estimates have been extensively presented in the method section (supplementary material). This period is essentially the time interval characterized by a warming trend during which the deposit formed according to radiocarbon dates (from ca. 14000 to ca. 7000, Fig S6).

We acknowledge that lack of clarity and we have changed the text so that it now provides more information about this time period (line 184-185).

- line 176, same: the land oce export of 4.5 TgC/yr: in which time period?

Same time period as reported above. Here we have simply divided the buried OC just mentioned for 7000 years to obtain the average annual OC flux (line 185).

- line 204: CO₂ rise during deglaciation is 80 ppmv (not 90 ppmv as stated) for Termination 1 (this is different/higher for other terminations, but since this paper is on the end of Termination 1 you should adjust this number.

We have changed the text accordingly (line 237)

Reviewer#2

This manuscript addresses a really interesting topic and puts forward a thought-provoking argument for permafrost carbon destabilization as an important source of atmospheric carbon dioxide during post-glacial warming. Overall, I find the data put forward to be of a high quality and the manuscript is certainly going to stimulate a lot of discussion and I therefore believe will be of high enough impact for Nature Communications. The manuscript is well-written, clearly organized, in the appropriate format and all the data put forward is well supported. That said I have some major concerns that need addressing prior to potential publication.

Firstly, everything is really based off of one piston core, I am well aware of the amount of work that goes into this kind of study but I have serious concerns about how representative data from one core is for scaling up from - can the authors provide any further data to back up the representative nature of this core or other similar data from other cores?

It is correct that the new strategically-located piston core is characterized to a much greater detail than ever done before for a Holocene sediment record in the Arctic. However, the spatial extrapolation is benefitting greatly from and leveraging off a larger set of piston cores across the Laptev Sea. These earlier heroic efforts (Bauch et al., 2001; Stein and Fahl, 2000; Taldenkova et al., 2005) combine with the new record (adding molecular and compound-specific isotope dimensions) to now allow estimation of the burial specifically of the terrestrial-C fraction in the Laptev Sea during the last deglaciation. For three of the earlier published records, bulk-C stable isotope data are available (Mueller-Lupp et al., 2000). In the text (line 179-181, version with track changes) we have mentioned that the stable carbon isotope composition of PC23 is consistent with the depleted signature of these three cores which indicates a common land-derived OC origin. To make it more explicit we have added a figure (Fig. S7) in the Supplementary Information to show the actual $\delta^{13}\text{C}$ data. It is also important to point out again that our core has been collected in the middle of the other sediment records (line 114-120).

Secondly, in lines 190-196 the authors describe how a lot of the CO₂ flux from the thawed permafrost would have happened from soils and throughout the fluvial network (the latter has been the focus of recent papers, e.g. Mann et al., 2015: doi:10.1038/ncomms8856; Drake et al., 2015: doi: 10.1073/pnas.1511705112) and this has been shown to be driven through leaching to DOC and then microbial utilization with likely significant outgassing in the

aquatic pathway. This should be included in here to make it clear how much C was on the move at this time as what was thawed and buried in the core represents the fraction preserved (i.e. a small proportion of what was likely thawed).

We agree with this notion and have further elaborated on this in the text to make it more clear how degradation occurs upon thawing followed by transport to the Laptev Sea, and now citing Mann et al and Drake et al papers (line 211-216)

Finally, in lines 199-203 the authors provide literature data for how much permafrost carbon can be mobilized and the size of the Lena watershed but do not provide an estimate for how much C could have been mobilized - this is a critical proof of concept that should be included. Basically, how much C could have been mobilized and then firstly for the Lena and then other Siberian watersheds how significant is the total C mobilized in comparison to the increase of atmospheric CO₂ (90ppmv) - basically can you show that enough C was mobilized to drive what is hypothesized?

The contemporary PF-C stock in northern high-latitude regions is about 1300 Pg C (Hugelius et al., 2014). Modelling studies suggests that the PF-C stock before the last deglaciation was even higher (i.e. 700 PgC higher) (Ciais et al., 2012). The increase of the atmospheric C during last deglaciation was about 190 PgC (Ciais et al., 2012). Taken together, these figures prove that there was definitely enough carbon in high-latitude soils to justify a putative role exerted by permafrost thawing/degradation on the post-glacial CO₂ rise.

We have changed the text to make the reader more aware about the size of the PF-C pool in respect to the net increase of the atmospheric C pool since the LGM (line 58-59 & 64)

Reviewer#3

Tesi and co-authors present data from a new sediment core retrieved from the Laptev Shelf, more specifically from a paleochannel of river Lena at around 50 m water depth (unfortunately I could not find information in the manuscript on the exact water depth from where the core was raised).

The water depth is in the Supplementary Information in the “Sampling and sediment core handling” section. We also added the water depth in the main text to make it more visible (line 94 & 251)

The sediment record covers approximately the past 12 kyr and is characterized by a drastic change in sedimentation rate from very high rates before approximately 11.5 kyr to moderately high rates afterwards. Next to the radiocarbon data used for the age model, the data obtained for this core comprise organic carbon contents, $\delta^{13}\text{C}$ of organic carbon, δD of long-chain n-alkanes, lignin and cutin concentrations, as well as $\Delta^{14}\text{C}$ of total organic carbon. The authors interpret their data, and a compilation of published data from the region, as records of organic matter remobilization from thawing permafrost deposits and suggest that the process strongly impacted atmospheric CO₂ levels.

This is a timely study investigating possible sources of CO₂ released during the deglaciation. Thawing terrestrial permafrost deposits have recently gained increasing attention due to their potential to explain part of the deglacial rise in atmospheric greenhouse gases, as well

as due to the potential effect resulting from anthropogenic climate change causing Arctic warming. The records presented here are of good quality and clearly indicate deglacial accumulation of terrigenous organic matter. However, the finding of strongly increased accumulation of terrigenous organic matter in sediments of the Laptev Shelf and slope, namely in paleochannels of the rivers, is not new. There are several publications discussing these deposits in context with deglacial sea-level rise (e.g., Bauch et al., 2001, cited in this study). In contrast, Tesi et al. argue, based on a mixing model using $\delta^{13}\text{C}$, $\Delta^{14}\text{C}$ and lignin concentrations, that "undoubtedly... (permafrost-carbon) was supplied from the watershed via surface water runoff rather than thermal collapse or erosion of ice-rich deposits."

The contrasting view of Tesi et al. relative to that of the authors of previous studies, namely that the authors of this study consider coastal erosion of ice-rich yedoma type deposits during flooding of previously exposed shelves a less important process, is backed up with the results of a mixing model relying on end-members defined from modern observations. While care is taken to correct $\Delta^{14}\text{C}$ of organic matter samples from the sediment core for radioactive decay that occurred since deposition, the end-member $\Delta^{14}\text{C}$ values apparently have not been corrected for decay. This would, however, be necessary for the ice-complex end-member, which today represents a glacial relict deposit and during the last glacial and deglaciation likely has formed the majority of deposits in the study area.

We are grateful to the reviewer for alerting us about the changing reservoir age of the Ice-Complex Deposit (ICD) PF/C, which is a correct technical point that we had overlooked. We have now re-run new Monte Carlo simulations to account for the ^{14}C decay of the ice-complex deposit ICD endmember as suggested by the reviewer. To do so, in the new model each MC simulation was run with a different $\Delta^{14}\text{C}$ value of ICD depending on its age (i.e progressively less depleted with increasing sediment age). The correction is based on Eq. 4 in the Supplementary Information. We also modified Fig. 3 and the Supplementary Material (see section "4.2 Endmember definition") based on this comment.

The new results do not change any conclusion. The fraction of the organic carbon in the massive early Holocene carbon deposit that are estimated to stem from washout of active layer topsoil-permafrost-C (f_{AL}) changed from 0.71 ± 0.05 in the first Monte Carlo simulation to 0.68 ± 0.07 in the now improved simulation. That the change is small is reasonable considering that (i) the MC simulation is affected by all variables (and associated uncertainties) and (ii) that the actual change in $\Delta^{14}\text{C}$ after the correction was relatively small due to the exponential relationship between age and ^{14}C abundance. For example, a $\Delta^{14}\text{C}$ value of -960‰ (about 26,000 years) corrected for 11,500 years equals -842‰ .

The end-member defined for top-soil relies on (arguably scarce) data from modern active layer soils in the Lena catchment and delta, which today consists of deposits fundamentally different from those probably prevalent in the region during the glacial, i.e., the ice-complex deposits.

The contemporary vegetation is surely different in respect to a glacial period as suggested by the reviewer because of the different climate. However, the modern active-layer is relatively similar to the ICD in terms of molecular fingerprint/biomarkers (lignin phenols) and $\delta^{13}\text{C}$ (while extremely different for the radiocarbon signature). This is reasonable considering that

both LGM and Holocene vegetation originally derive from ^{13}C depleted vascular plant material. Therefore we do not expect large variability of the active-layer in terms of $\delta^{13}\text{C}$ and lignin content over time.

Another relevant element to consider here is our statistical approach with relays on the Monte Carlo simulation. The Monte Carlo allows us to analyze the uncertainty propagation with the objective to determine how random variation and endmember uncertainty affects the sensitivity of our mixing model. Taken together, we believe that given the data available in the literature, this is a reasonable/best estimate at present.

Moreover, I feel that the authors do not fully exploit the potential of their data. They have obtained lignin phenol concentrations for the core. Concentration ratios of individual phenols can be used to apportion vegetation and plant tissue types, which are expected to differ between the southern reaches of the Lena catchment and the northern tundra. Yet, only lignin concentrations are presented (in mg/g OC, which is an uncommon unit in the literature dealing with lignin; in my view the more common unit should be used to assure easy comparison with other published data, i.e., $\lambda\text{-8}$ expressed in mg lignin phenols/100 mg OC). It would be very interesting to see whether the lignin phenol signature supports the assertion of large amounts of organic matter being sourced "from the watershed", which I would assume to encompass also the southern parts of the catchment, where the deglacial warming and permafrost thawing must have been strongest.

We have changed the unit according to the reviewer's suggestion from mg/g OC to mg/100mg OC (Fig. S4b). Furthermore, in Fig. S4b we added the mean lignin values of ICD and PF-C as additional reference for the reader. The overall picture confirms the large terrigenous influence.

Tesi and co-authors furthermore argue that the stable hydrogen isotopic composition of n-alkanes in their core supports an origin from the catchment of the river, as the δD values in the rapidly accumulating late deglacial sediments are similar to values found in a loess deposit 1200 km upstream from the current Lena Delta. They state that the hydrogen isotopes indicate a source from plants growing under cold climate. However, no data exist on the δD signature of alkanes from ice-complex deposits, which were likely prevalent on the flooded shelf. It is conceivable that those alkanes would also carry a δD signature indicative of cold conditions. So the δD data alone do not support an exclusion of coastal erosion of deposits on the flooded shelf as source for the material accumulated in the channel.

This may be a misunderstanding as there are no claims based on δD -alkane of the relative source contribution from upland/inland catchment vs coastal ICD. The δD -alkane values are used here only to show that the C was produced during a cold period as specified in the text (line 159-160). So, the manuscript is consistent with the reviewer comment that the δD -alkane alone does not exclude a large contribution from erosion of coastal ICD (it is the ^{14}C -OC that constrains this). In other words, the deposited organic matter was once actual PF-C which eventually thawed out in response to the warming climate. Given that the Lena river extends southwards for 20 latitudinal degrees, this material might hypothetically also be sourced from the unfrozen southernmost Lena watershed. However, the δD -alkane data strongly suggests that the carbon once was PF-C formed in a cold environment.

In conclusion, as the reviewer said, a depleted δD for the ICD can be expected as well because this deposit predominantly formed during a really cold period. However, the actual OC source in the paper is exclusively based on the 3-endmember mixing model and its Monte Carlo simulation. We have further elaborated the text to make our overarching message clearer (line 166-167)

In the second part of the manuscript, an attempt is made to estimate the amount of carbon mobilization during deglacial permafrost thaw and sea-level rise. This is done by extrapolating the data from the core studied here and from other published records to the entire Laptev Shelf area. However, considering that the cores were predominantly and deliberately taken in paleochannels of the rivers, as these locations offer high-resolution sediment sequences resulting from predominant deposition in these paleo-depressions, upscaling accumulation rates found in these settings must result in vast overestimations.

The combined Holocene-scale records used for the upscaling are from diverse locations and depositional settings throughout the Laptev Sea (see Fig. 1). Some records are certainly from paleo-channels, as these settings are ubiquitous large-scale features of the current Laptev shelf and thus should also be included. While even greater coverage (more data) certainly is always helpful, we believe that this first attempt at upscaling is timely and justified – it gives a first iteration of the scale of this massive permafrost-carbon remobilization to the marginal sea receptor during an early Holocene rapid warming event.

Overall, we nevertheless agree with the statement of the reviewer and have changed the main text to acknowledge the potential bias derived from the dataset used for the interpolation (line 185-190). Within the discussion, we now also point out that the greatest sediment thicknesses are located in the outer-shelf (line 188-190). Based on the latest IBACAO bathymetry these sites do not appear to be in major paleo-channel environments. These outer-shelf cores, which likely recorded the initial/mid-phase of the deglaciation, have comparatively more weight on the overall upscaling. All of this is now added to the revised manuscript (line 185-190).

Lastly, it would be desirable to provide the exact results of the end-member modelling, perhaps in a table in the supplement.

The Table with the Monte Carlo results is now part of the Supplementary Information (Table S5).

References

- Bartolomé, M., Moreno, A., Sancho, C., Stoll, H. M., Cacho, I., Spötl, C., Belmonte, Á., Edwards, R. L., Cheng, H., and Hellstrom, J. C., 2015, Hydrological change in Southern Europe responding to increasing North Atlantic overturning during Greenland Stadial 1: Proceedings of the National Academy of Sciences, v. 112, no. 21, p. 6568-6572.
- Bauch, H. A., Mueller-Lupp, T., Taldenkova, E., Spielhagen, R. F., Kassens, H., Grootes, P. M., Thiede, J., Heinemeier, J., and Petryashov, V., 2001, Chronology of the Holocene transgression at the North Siberian margin: Global and Planetary Change, v. 31, no. 1, p. 125-139.

- Brook, E. J., Harder, S., Severinghaus, J., Steig, E. J., and Sucher, C. M., 2000, On the origin and timing of rapid changes in atmospheric methane during the last glacial period: *Global Biogeochemical Cycles*, v. 14, no. 2, p. 559-572.
- Ciais, P., Tagliabue, A., Cuntz, M., Bopp, L., Scholze, M., Hoffmann, G., Lourantou, A., Harrison, S. P., Prentice, I., and Kelley, D., 2012, Large inert carbon pool in the terrestrial biosphere during the Last Glacial Maximum: *Nature Geoscience*, v. 5, no. 1, p. 74-79.
- Drake, T. W., Wickland, K. P., Spencer, R. G. M., McKnight, D. M., and Striegl, R. G., 2015, Ancient low-molecular-weight organic acids in permafrost fuel rapid carbon dioxide production upon thaw: *Proceedings of the National Academy of Sciences*, v. 112, no. 45, p. 13946-13951.
- Dutta, K., Schuur, E., Neff, J., and Zimov, S., 2006, Potential carbon release from permafrost soils of Northeastern Siberia: *Global Change Biology*, v. 12, no. 12, p. 2336-2351.
- Hugelius, G., Strauss, J., Zubrzycki, S., Harden, J. W., Schuur, E., Ping, C.-L., Schirmer, L., Grosse, G., Michaelson, G. J., and Koven, C. D., 2014, Estimated stocks of circumpolar permafrost carbon with quantified uncertainty ranges and identified data gaps: *Biogeosciences*, v. 11, no. 23, p. 6573-6593.
- Knoblauch, C., Beer, C., Sosnin, A., Wagner, D., and Pfeiffer, E. M., 2013, Predicting long-term carbon mineralization and trace gas production from thawing permafrost of Northeast Siberia: *Global change biology*, v. 19, no. 4, p. 1160-1172.
- Lee, H., Schuur, E. A., Inglett, K. S., Lavoie, M., and Chanton, J. P., 2012, The rate of permafrost carbon release under aerobic and anaerobic conditions and its potential effects on climate: *Global Change Biology*, v. 18, no. 2, p. 515-527.
- Mann, P. J., Eglinton, T. I., McIntyre, C. P., Zimov, N., Davydova, A., Vonk, J. E., Holmes, R. M., and Spencer, R. G., 2015, Utilization of ancient permafrost carbon in headwaters of Arctic fluvial networks: *Nature communications*, v. 6.
- Mueller-Lupp, T., Bauch, H. A., Erlenkeuser, H., Hefter, J., Kassens, H., and Thiede, J., 2000, Changes in the deposition of terrestrial organic matter on the Laptev Sea shelf during the Holocene: evidence from stable carbon isotopes: *International Journal of Earth Sciences*, v. 89, no. 3, p. 563-568.
- Olefeldt, D., Turetsky, M. R., Crill, P. M., and McGuire, A. D., 2013, Environmental and physical controls on northern terrestrial methane emissions across permafrost zones: *Global Change Biology*, v. 19, no. 2, p. 589-603.
- Ruth, U., Wagenbach, D., Steffensen, J. P., and Bigler, M., 2003, Continuous record of microparticle concentration and size distribution in the central Greenland NGRIP ice core during the last glacial period: *Journal of Geophysical Research: Atmospheres*, v. 108, no. D3.
- Schuur, E. A., Vogel, J. G., Crummer, K. G., Lee, H., Sickman, J. O., and Osterkamp, T. E., 2009, The effect of permafrost thaw on old carbon release and net carbon exchange from tundra: *Nature*, v. 459, no. 7246, p. 556-559.
- Smith, L., MacDonald, G., Velichko, A., Beilman, D., Borisova, O., Frey, K., Kremenetski, K., and Sheng, Y., 2004, Siberian peatlands a net carbon sink and global methane source since the early Holocene: *Science*, v. 303, no. 5656, p. 353-356.
- Steffensen, J. P., Andersen, K. K., Bigler, M., Clausen, H. B., Dahl-Jensen, D., Fischer, H., Goto-Azuma, K., Hansson, M., Johnsen, S. J., and Jouzel, J., 2008, High-resolution Greenland ice core data show abrupt climate change happens in few years: *Science*, v. 321, no. 5889, p. 680-684.
- Stein, R., and Fahl, K., 2000, Holocene accumulation of organic carbon at the Laptev Sea continental margin (Arctic Ocean): sources, pathways, and sinks: *Geo-Marine Letters*, v. 20, no. 1, p. 27-36.

- Taldenkova, E., Bauch, H. A., Stepanova, A., Dem'yankov, S., and Ovsepyan, A., 2005, Last postglacial environmental evolution of the Laptev Sea shelf as reflected in molluscan, ostracodal, and foraminiferal faunas: *Global and Planetary Change*, v. 48, no. 1, p. 223-251.
- Xue, K., M. Yuan, M., J. Shi, Z., Qin, Y., Deng, Y., Cheng, L., Wu, L., He, Z., Van Nostrand, J. D., Bracho, R., Natali, S., Schuur, E. A. G., Luo, C., Konstantinidis, K. T., Wang, Q., Cole, J. R., Tiedje, J. M., Luo, Y., and Zhou, J., 2016, Tundra soil carbon is vulnerable to rapid microbial decomposition under climate warming: *Nature Clim. Change*, v. advance online publication.

Reviewers' comments:

Reviewer #1 (Remarks to the Author):

Review on

Title: Massive remobilization of permafrost carbon during post-glacial warming

Authors: T. Tesi et al.

2nd submission to Nature Communications

Today: 08 June 2016

Since this the 2nd round of review, I refuse following the suggested order of a review and focus only on still open issues:

I found the paper has significantly improved and I have only one major comment, a few moderate comments, and some technical corrections:

Major:

In response to main comment 2 of review 1 they now discuss $\delta^{14}\text{C}_{\text{CH}_4}$ as published by Petrenko et al 2009 in Science. However, they cite results from that paper as $\delta^{14}\text{C}$ values of -100 to -200 permil at the end of the YD. However, I can not find these numbers in the Petrenko paper, there the $\delta^{14}\text{C}_{\text{CH}_4}$ values (corrected for cosmogenic $\delta^{14}\text{C}$, red diamonds in Fig 1 of Petrenko) are around +200 permil at the end of the YD, and slightly lower (maybe 0-+200 permil including the uncertainties) during the transition in the Preboreal and the onset of the Holocene, but never below 0 permil. This needs clarification / correction. If once clarified what would this imply for the interpretation based on these data included in the revised version? I acknowledge, that Petrenko labeled the numbers $\delta^{14}\text{C}_{\text{CH}_4}$ (in permil), not $\delta^{14}\text{C}_{\text{CH}_4}$. So if there is some hidden transformation done by the authors from one into the other, this needs to be layed out in the draft.

Moderate:

1) Around line 68 it is argued that still no deep ocean reservoir with $\delta^{14}\text{C}$ -depleted carbon has been found, which is why one of the hypothesis (upwelling of poorly ventilated abyssal waters) explaining CO_2 rise during deglaciation still lacks some support. However, since a month or so this

is not the case anymore, since in Ronge et al 2016 (NC) a glacial carbon pool in the Pacific with ^{14}C depletion has been found, although its extent is still poorly constrained and a model-based interpretation of consequences for CO_2 is missing. So, this discussion needs some revision based on this new paper, citation below.

Ronge et al. Radiocarbon constraints on the extent and evolution of the Pacific glacial carbon pool
Nature Communications, 2016, 7, 11487; doi: 10.1038/ncomms11487

2) Around lines 160-167 it is argued that "the $\delta^2\text{H}$ signature ... indicates that the carbon deposited during the YD-PB transition is PF-C originally formed during a cold period in the northern permafrost domain...".

- Why is the depleted $\delta^2\text{H}$ signature an indicator that the carbon originates from northern permafrost, and not from soil OC from southernmost regions? I think a key argument is missing here.

- How can carbon be formed in a cold period in a permafrost domain? Do you mean photosynthetically produced? Or anything else, if so, please elaborate how this might happen. Maybe rewording might clarify here.

Technicals:

- no citation in abstract in this journal

- lines 63ff: CO_2 rise from 190 to 280 ppmv = 90 ppmv during the last interglacial transition is wrong. Here they cite Ciais et al 2012 NG (ref 2), which compares LGM with pre-industrial. During Termination I the rise in CO_2 is 80 ppmv from 190 to 270 ppmv, the remaining rise from 270 to 280 ppmv until the pre-industrial is happening in the Holocene. The correct citation for the CO_2 rise is either Monnin et al 2001 (their ref 6), or even better, because most recently published in higher resolution is:

Marcott et al., Centennial Scale Changes in the Global Carbon Cycle During the Last Deglaciation
Nature, 2014, 514, 616-619

- lines 217-218: 2x "likely" in one sentence.

- line 237: here CO_2 rises by 80 ppmv during the deglaciation, which is correct, but different than what is written in the intro (lines 63ff).

- References: For online papers, e.g. from Nature Communications, the paper numbers are missing (e.g. Kohler et al 2014, ref 1).

- Caption to Fig 4: "Upper x-axis shows the core depth (cm)". This is not an upper x-axis, but the legend of the color code.

- SI: "Figure S1" is called "Figure 1"

- SI: Section 5: data base is called "PANGAEA", not "PANGEA".

- SI-reference 11 is corrupted, only called "Bronk Ramsey C. (2010)".

Reviewer #2 (Remarks to the Author):

I previously reviewed this manuscript and feel like the authors have done a good job of including

my suggestions or answering my concerns. They have made the scaling potential clear and so I'm excited to see this paper hopefully move forward to publication. I think it will be a strong addition to the literature and cite much discussion.

Reviewer #3 (Remarks to the Author):

Tesi et al. present data from a sediment core recovered from a paleochannel of the river Lena in the Laptev Sea at approximately 56 m modern water depth. The data document the deposition of mainly terrigenous, ^{14}C depleted organic matter at high rates during an interval of approximately 1000 years at the end of the Younger Dryas and during the transition towards the Preboreal. The authors perform dual isotope modelling to estimate the source of the terrigenous organic matter based on end-members defined for the modern conditions in the study region. The results of this end-member modelling are taken as indication for warming induced massive discharge of terrestrial organic matter to the Laptev Sea during the late YD as the cause for the high accumulation rate event. In the second part of the manuscript these new data are compiled with published evidence for periods of high terrigenous organic matter accumulation in the region, and an attempt is made at estimating the total amount of terrigenous carbon released. The data are furthermore discussed in terms of their implications for potential greenhouse gas release from thawing permafrost.

The paper addresses the important issue of the role carbon release from thawing permafrost deposits in climate change. The record presented is interesting, but I don't think it is sufficient to back up all the inferences made. First of all, it is too short to cover the period of interest. The core only reaches back to approximately 11.8 ka BP. In order to fully document the role of OM release from thawing permafrost for the deglacial CO_2 rise, the focus should be put on a full deglacial record including those periods, for which specifically rapid release of carbon from land has been brought forward as a potential mechanisms, i.e., the rapid rise in atmospheric CO_2 at 14.6 ka BP. The record presented here provides evidence for a period of rapidly increased accumulation, which in itself is not new for the region. There are many published records showing such a drastic increase in sediment accumulation (see Bauch et al., 2001, Global and Planetary Change, cited by the authors but only for using the data on high OM accumulation, not in context of the role of sea-level rise). The added value of this new record is that next to accumulation rates and bulk OM contents, bulk ^{14}C values are provided along with terrigenous biomarker concentrations and stable hydrogen isotopes of n-alkanes.

While I have no doubts that the observed deposit is really evidence for the mobilization of terrigenous carbon, I don't think that the inferences made by the authors are correct. They regard the role of sea-level rise as minor for the deposition of this layer, but this is not discussed extensively. As expressed in my review of the first version of this manuscript, I am not convinced that the process responsible for the high accumulation rate of terrigenous organic matter can be identified as warming of northern permafrost deposits as suggested by the authors. The reasons for this are:

- 1) The period of increased terrigenous organic matter accumulation in the core starts well before the warming of high northern latitudes as observed in Greenland (assuming that Greenland temperature is an adequate approximation of the temperature development in the Lena watershed, which in itself is rather controversial). There is evidence that during the Younger Dryas, there was severe cooling in Beringia (Meyer et al., 2010, GRL), rendering the scenario of warming induced C mobilization from the Lena watershed during the YD rather implausible. In the rebuttal letter, in reply to a comment along the same lines made by reviewer #1, the great latitudinal extent of the Lena watershed is brought forward in connection with records of dust in Asian desert, which is taken as evidence for increased humidity and resulting land-ocean C transfer. At the same time, the authors state that the primary origin of the OM found in the marine deposit must be from northern sources. In order to provide compelling evidence for warming

induced OM export, a regional continental temperature record would be needed.

Besides, as also already suggested in my first review, the authors should make use of the full information contained within the lignin data, i.e., the possibility to assess the vegetation type in the source area by studying the relative abundance of the individual phenols released from the lignin polymer. These data are not discussed, but they potentially could provide valuable clues on the origin of the terrestrial organic matter.

2) The high accumulation rate organic rich deposits known from the Laptev Sea have previously been interpreted as being related to sea-level rise. In the supplement, Tesi et al. also refer to these deposits as "transgressive". The period of high terrigenous OM accumulation at the study site at 56 m water depth occurs exactly when sea-level reconstructions from the area show that the shoreline was at approximately the position where the core was taken (see e.g., Klemann et al., 2015 Figure 3). It is therefore conceivable, that the primary cause for high OM deposition at the core site is indeed the sea-level rise. The fact that the core ends at roughly 11.8 ka BP may be indeed be related to only terrigenous sandy deposits being below this level; of course this is speculative.

3) The end member model relies on isotopic values defined for the modern situation. As the authors are well aware, the glacial and deglacial environment in the northern Lena watershed must have been different from today. In the rebuttal, the authors argue that at present, the $\delta^{13}\text{C}$ and lignin composition in ICD is similar to that of Holocene deposits active layers. Regardless of the validity of this further assumption, the results from end-member modelling should be treated with caution. As is, they are the only evidence for the inferred active layer source of the massive amounts of carbon translocated to the depositional sites in the Laptev Sea. No evidence is provided for the inferred massive increase in discharge, which should have left a trace in the record as it would impact, e.g., salinity. Stable isotope records of planktic foraminifera or of plankton biomarkers could be used to show discharge induced changes in salinity. To my knowledge, the only available record showing such massively increased discharge of the Lena dates it to 13 ka BP (Spielhagen et al, 2005, Global and Planetary Change). Furthermore, no evidence is provided against sea-level rise as the primary cause for the high accumulation rate event.

4) The erosive process causing particulate organic matter discharge from permafrost areas to the ocean is not fully understood and not satisfactorily explained in the manuscript. At present, most OM is released during the ice break-up in spring, when ice blocks effectively scrape off the terrestrial cliff deposits along the river banks, mainly of the delta (see Fedorova et al., 2015). I don't see how this process would drastically change in response to warming.

Minor comments:

Line 68/69: It is not quite true that a ^{14}C -depleted deep ocean reservoir has not been found. The cited reference is 5 years old. More recent papers have shown the existence of large ^{14}C depleted C pools in the intermediate depths of the Southern Ocean and South Pacific (e.g., Skinner et al., 2014, 2015, Ronge et al., 2016)

Line 101: I believe the reference given here for ^{210}Pb and ^{137}Cs dates for the "Lena Delta" may be wrong. Moreover, I think it should be more appropriately referred to the Lena Pro-Delta, as certainly accumulation rates of marine sediments are referred to.

Line 108: Again, the term "Lena delta" is misleading here.

**Author responses to review comments on Nature Comm. ms NCOMMS-16-03685A
"Massive remobilization of permafrost carbon during post-glacial warming"**

We appreciate all the constructive comments on the revised version of our manuscript NCOMMS-16-03685A and glad to see that rev#2 considers our work ready for publication. Rev#1 acknowledges that the paper “*has significantly improved*” but a few clarifications and small changes in the text are still necessary prior to publication. As pointed out by the Associate Editor, Rev#3 still holds some reservations on the mechanism/source of the deglacial high fluxes of terrestrial OC and suggests that sea level ingression may be the most likely candidate.

Rev#3 provided also several constructive suggestions on how to further test this. Among these, we have further dug into the chemical biomarker fingerprint to look for clues about the source of the massive terrestrial OC input. This new information on the detailed lignin composition adds to the other data (e.g. mixing model calculations and hydrogen-isotope composition of wax lipids) and further supports that the massive carbon input largely stemmed from a watershed washout signal rather than due to sea-level induced coastal erosion (i.e., coastal Ice Complex Deposits have another distinct biomarker fingerprint compared to our sediment record). This Rev#3-stimulated new information is now part of the revised main text (line 153-161 and new Fig. 5).

As for the first revision, the line numbers below refers to the latest revised version. The following document has been organized such that first reviewer comments are given in italic directly followed by our detailed response in regular colored blue font.

Reviewer #1 (Remarks to the Author):*I found the paper has signifantly improved and I have only one major comment, a few moderate comments, and some technical corrections:
Major:*

In response to main comment 2 of review 1 they now discuss D14CH4 as published by Petrenko et al 2009 in Science. However, they cite results from that paper as 14C values of -100 to -200 permil at the end of the YD. However, I can not find these numbers in the Petrenko paper, there the 14CH4 values (corrected for cosmogenic 14C, red diamonds in Fig 1 of Petrenko) are around +200 permil at the end of the YD, and slightly lower (maybe 0-+200 permil including the uncertainties) during the transition in the Preboreal and the onset of the Holocene, but never below 0 permil. This needs clarification / correction. If once clarified what would this imply for the interpretation based on these data included in the revised version? I acknowledge, that Petrenko labeled the numbers 14CH4 (in permil), not D14CH4. So if there is some hidden transformation done by the authors from one into the other, this needs to be layed out in the draft.

We acknowledge this inaccuracy and changed the text accordingly (line 256-263). As pointed out by the reviewer, the cosmogenic-corrected (and age-corrected) radiocarbon values of CH4 presented in

Petrenko et al 2009 range between +100 and 200 per mil. Comparison with our data (after correcting for the age with Intcal13 and the decay until the YD) reveals a difference of ca. 250 permil (i.e., more depleted compared the $^{14}\text{CH}_4$ signature). Petrenko et al have also recently revised their calculations¹ using a different method which further confirmed contemporaneous methane sources likely from wetlands.

Moderate:

1) Around line 68 it is argued that still no deep ocean reservoir with 14C-depleted carbon has been found, which is why one of the hypothesis (upwelling of poorly ventilated abyssal waters) explaining CO₂ rise during deglaciation still lacks some support. However, since a month or so this is not the case anymore, since in Ronge et al 2016 (NC) a glacial carbon pool in the Pacific with 14C depletion has been found, although its extend is still poorly constrained and a model-based interpretation of consequences for CO₂ is missing. So, this discussion needs some revision based on this new paper, citation below.

Ronge et al. Radiocarbon constraints on the extent and evolution of the Pacific glacial carbon pool *Nature Communications*, 2016, 7, 11487; doi: 10.1038/ncomms11487

We thank the reviewer for pointing out the recent publication by Ronge et al. We have changed the text based on this comment (line 67-69). We also would like to point out that in the current study we do not reject the ocean hypothesis *per se* but, as stated in the text (line 264-266), we envision a combination of multiple processes (line 267, "the emerging picture suggests that a combination of processes must have been operating") based on collective evidence found in the literature.

2) Around lines 160-167 it is argued that "the d₂H signature ... indicates that the carbon deposited during the YD-PB transition is PF-C originally formed during a cold period in the northern permafrost domain...".

- Why is the depleted d₂H signature an indicato

REVIEWERS' COMMENTS:

Reviewer #1 (Remarks to the Author):

I am satisfied with the responses and changes following my comments. However, I believe the concern of reviewer 3 needs to be addressed in detail to come to a final version here. If changes and responses in that direction are sufficient is something which only reviewer 3 can decide.

Reviewer #3 (Remarks to the Author):

In my opinion, the paper has significantly improved over the previous versions. In particular, I am delighted to finally see the detailed lignin data, which, as the authors seem to agree, add extremely valuable information and backup for the interpretation. I am also happy to see additional evidence for the climatic evolution in the Lena drainage area, as well as indications for

changes in surface water salinity in the Laptev Sea. Based on this new evidence, I agree that the records presented can be interpreted in the way the authors suggest.

I only have a few minor, mainly editorial, points:

In line 197/198, it would be good to add over which time period the increased annual land-to-ocean export occurred.

In line 254, I would replace "magnitude" by "amount"

In line 257, it is stated that the $\Delta^{14}\text{C}$ signature of remobilized active layer permafrost (in itself a term that is debatable, as "permafrost" refers to soil or ground that remains frozen over two or more consecutive years, while the active layer is on top of this and thaws in summer; perhaps one could say more precisely: carbon/organic matter remobilized from the active layer of permafrost affected areas) is "relatively more depleted (ca. 250‰) than... CH_4 ". I think this is misleading. From the table presented in the supplement, I calculate an average decay-corrected $\Delta^{14}\text{C}$ of -264‰ for all samples between 28.5 and 233.5 m core depth, which is approximately 400 ‰ more depleted than the age-corrected CH_4 values of +150‰. This needs to be clarified.

Furthermore, there are several errors in the reference table. In particular, in the new figure 5, data from two papers are cited, but the attributions are wrong. As is, the data are attributed to reference 21 and 23. They should, however, correctly be attributed to Winterfeld et al., Characterization of particulate organic matter in the Lena River delta and adjacent nearshore zone, NE Siberia – Part 2: Lignin-derived phenol compositions. Biogeosciences 12, 2261-2283, doi: 10.594/bg-12-2261-2015, 2015, and Tesi et al., Comparison and fate of terrigenous organic matter along the Arctic land-ocean continuum....GCA, 2014, listed as ref. 27 in the current list.

In line 197/198, it would be good to add over which time period the increased annual land-to-ocean export occurred.

The time interval was added as asked by the reviewer (line 193)

In line 254, I would replace "magnitude" by "amount"

Replaced as suggested

In line 257, it is stated that the $\Delta^{14}\text{C}$ signature of remobilized active layer permafrost (in itself a term that is debatable, as "permafrost" refers to soil or ground that remains frozen over two or more consecutive years, while the active layer is on top of this and thaws in summer; perhaps one could say more precisely: carbon/organic matter remobilized from the active layer of permafrost affected areas) is "relatively more depleted (ca. 250‰) than... CH_4 ". I think this is misleading. From the table presented in the supplement, I calculate an average decay-corrected $\Delta^{14}\text{C}$ of -264‰ for all samples between 28.5 and 233.5 m core depth, which is approximately 400 ‰ more depleted than the age-corrected CH_4 values of +150‰. This needs to be clarified.

The reviewer is right. The -250‰ at line 257 referred to the decay-corrected average radiocarbon value of the sedimentary OC of PC23 (as also shown in Fig. 3). As the age corrected radiocarbon

value of CH₄ during the YD-PB transition is +150‰, the actual difference is 400 ‰. The text was changed accordingly.

Furthermore, there are several errors in the reference table. In particular, in the new figure 5, data from two papers are cited, but the attributions are wrong. As is, the data are attributed to reference 21 and 23. They should, however, correctly be attributed to Winterfeld et al., Characterization of particulate organic matter in the Lena River delta and adjacent nearshore zone, NE Siberia – Part 2: Lignin-derived phenol compositions. *Biogeosciences* 12, 2261-2283, doi: 10.5194/bg-12-2261-2015, 2015, and Tesi et al., Comparison and fate of terrigenous organic matter along the Arctic land-ocean continuum....*GCA*, 2014, listed as ref. 27 in the current list.

This is another oversight which was corrected. We have double checked text and captions.